# Iteratively Refined Early Interaction Alignment for Subgraph Matching based Graph Retrieval

**Ashwin Ramachandran**[1*]    **Vaibhav Raj**[2*]    **Indrayumna Roy**[2]
**Soumen Chakrabarti**[2]    **Abir De**[2]
[1]UC San Diego    [2]IIT Bombay
ashwinramg@ucsd.edu
{vaibhavraj, indraroy15, soumen, abir}@cse.iitb.ac.in

## Abstract

Graph retrieval based on subgraph isomorphism has several real-world applications such as scene graph retrieval, molecular fingerprint detection and circuit design. Roy et al. [35] proposed IsoNet, a late interaction model for subgraph matching, which first computes the node and edge embeddings of each graph independently of paired graph and then computes a trainable alignment map. Here, we present IsoNet++, an early interaction graph neural network (GNN), based on several technical innovations. First, we compute embeddings of all nodes by passing messages within and across the two input graphs, guided by an *injective alignment* between their nodes. Second, we update this alignment in a lazy fashion over multiple *rounds*. Within each round, we run a layerwise GNN from scratch, based on the current state of the alignment. After the completion of one round of GNN, we use the last-layer embeddings to update the alignments, and proceed to the next round. Third, IsoNet++ incorporates a novel notion of node-pair partner interaction. Traditional early interaction computes attention between a node and its potential partners in the other graph, the attention then controlling messages passed across graphs. In contrast, we consider *node pairs* (not single nodes) as potential partners. Existence of an edge between the nodes in one graph and non-existence in the other provide vital signals for refining the alignment. Our experiments on several datasets show that the alignments get progressively refined with successive rounds, resulting in significantly better retrieval performance than existing methods. We demonstrate that all three innovations contribute to the enhanced accuracy. Our code and datasets are publicly available at https://github.com/structlearning/isonetpp.

## 1   Introduction

In graph retrieval based on subgraph isomorphism, the goal is to identify a subset of graphs from a corpus, denoted $\{G_c\}$, wherein each retrieved graph contains a subgraph isomorphic to a given query graph $G_q$. Numerous real-life applications, *e.g.*, molecular fingerprint detection [6], scene graph retrieval [16], circuit design [29] and frequent subgraph mining [43], can be formulated using subgraph isomorphism. Akin to other retrieval systems, the key challenge is to efficiently score corpus graphs against queries.

Recent work on neural graph retrieval [1, 2, 11, 22, 23, 35, 31, 46] has shown significant promise. Among them, Lou et al. [23, Neuromatch] and Roy et al. [35, IsoNet] focus specifically on subgraph isomorphism. They employ graph neural networks (GNNs) to obtain embeddings of query and corpus graphs and compute the relevance score using a form of order embedding [39]. In addition, IsoNet also approximates an *injective alignment* between the query and corpus graphs. These two models operate in a *late interaction* paradigm, where the representations of the query and corpus graphs are

---

*Equal contribution. Ashwin Ramachandran did this work while at IIT Bombay.

38th Conference on Neural Information Processing Systems (NeurIPS 2024).

computed independent of each other. In contrast, GMN [22] is a powerful *early interaction* network for graph matching, where GNNs running on $G_q$ and $G_c$ interact with each other at every layer.

Conventional wisdom suggests that early interaction is more accurate (even if slower) than late interaction, but GMN was outperformed by IsoNet. This is because of the following reasons. (1) GMN does not explicitly infer any alignment between $G_q$ and $G_c$. The graphs are encoded by two GNNs that interact with each other at every layer, mediated by attentions from each node in one graph on nodes in the other. These attentions are functions of node embeddings, so they change from layer to layer. While these attentions may be interpreted as approximate alignments, they induce at best non-injective mappings between nodes. (2) In principle, one wishes to propose a consistent alignment across all layers. However, GMN's attention based 'alignment' is updated in every layer. (3) GMN uses a standard GNN that is known to be an over-smoother [36, 40]. Due to this, the attention weights (which depend on the over-smoothed node representations) also suffer from oversmoothing. These limitations raise the possibility of a *third* approach based on early interaction networks, enabled with explicit alignment structures, that have the potential to outperform both GMN and IsoNet.

## 1.1 Our contributions

We present IsoNet++, an early interaction network for subgraph matching that maintains a chain of explicit, iteratively refined, injective, approximate alignments between the two graphs.

**Early interaction GNNs with alignment refinement**   We design early interaction networks for scoring graph pairs, that ensure the node embeddings of one graph are influenced by both its paired graph and the alignment map between them. In contrast to existing works, we model alignments as an explicit "data structure". An alignment can be defined between either nodes or edges, thus leading to two variants of our model: IsoNet++ (Node) and IsoNet++ (Edge). Within IsoNet++, we maintain a sequence of such alignments and refine them using GNNs acting on the two graphs. These alignments mediate the interaction between the two GNNs. In our work, we realize the alignment as a doubly stochastic approximation to a permutation matrix, which is an injective mapping by design.

**Eager or lazy alignment updates**   In our work, we view the updates to the alignment maps as a form of gradient-based updates in a specific quadratic assignment problem or asymmetric Gromov-Wasserstein (GW) distance minimization [30, 41]. The general form of IsoNet++ allows updates that proceed lockstep with GNN layers (*eager* layer-wise updates), but it also allows *lazy* updates. Specifically, IsoNet++ can perform $T$ *rounds* of updates to the alignment, each round including $K$ *layers* of GNN message passing. During each round, the alignment is held fixed across all propagation layers in GNN. At the end of each round, we update the alignment by feeding the node embeddings into a neural Gumbel-Sinkhorn soft permutation generator [10, 26, 37].

**Node-pair partner interaction between graphs**   The existing remedies to counter oversmoothing [8, 33, 40] entail extra computation; but they may be expensive in an early-interaction setting. Existing early interaction models like [22] perform *node partner interaction*; interactions are constrained to occur between a node and it's *partner*, the node in the paired graph aligned with it. Instead, we perform *node-pair partner* interaction; the interaction is expanded to include the *node-pairs* (or edges) in the paired graph that correspond to node-pairs containing the node. Consequently, the embedding of a node not only depends on nodes in the paired graph that align with it, but also captures signals from nodes in the paired graph that are aligned with its neighbors.

**Experiments**   The design components of IsoNet++ and their implications are subtle — we report on extensive experiments that tease out their effects. Our experiments on real world datasets show that, IsoNet++ outperforms several state-of-the-art methods for graph retrieval by a substantial margin. Moreover, our results suggest that capturing information directly from node-pair partners can improve representation learning, as compared to taking information only from node partner.

## 2 Preliminaries

**Notation**   Given graph $G = (V, E)$, we use nbr($u$) to denote the neighbors of a node $u \in V$. We use $u \to v$ to indicate a message flow from node $u$ to node $v$. Given a set of corpus graphs $C = \{G_c\}$ and a query graph $G_q$, we denote $y(G_c \mid G_q)$ as the binary relevance label of $G_c$ for $G_q$. Motivated by several real life applications like substructure search in molecular graphs [12], object search in scene graphs [16], and text entailment [20], we consider subgraph isomorphism to significantly influence the relevance label, similar to previous works [23, 35]. Specifically, $y(G_c \mid G_q) = 1$ when $G_q$ is a

subgraph of $G_c$, and 0 otherwise. We define $C_{q+} \subseteq C$ as the set of corpus graphs that are relevant to $G_q$ and set $C_{q-} = C \backslash C_{q+}$. Mildly overloading notation, we use $\boldsymbol{P}$ to indicate a 'hard' (0/1) permutation matrix or its 'soft' doubly-stochastic relaxation. $\mathcal{B}_n$ denotes the set of all $n \times n$ doubly stochastic matrices, and $\Pi_n$ denotes the set of all $n \times n$ permutation matrices.

**IsoNet [35]** Given a graph $G = (V, E)$, IsoNet uses a GNN, which initializes node representations $\{\boldsymbol{h}_0(u) : u \in V\}$ using node-local features. Then, messages are passed between neighboring nodes in $K$ *propagation layers*. In the $k$th layer, a node $u$ receives messages from its neighbors, aggregates them, and then combines the result with its state after the $(k-1)$th layer:

$$\boldsymbol{h}_k(u) = \text{comb}_\theta \left( \boldsymbol{h}_{k-1}(u), \sum_{v \in \text{nbr}(u)} \{\text{msg}_\theta(\boldsymbol{h}_{k-1}(u), \boldsymbol{h}_{k-1}(v))\} \right). \tag{1}$$

Here, $\text{msg}_\theta(\cdot)$ and $\text{comb}_\theta(\cdot, \cdot)$ are suitable networks with parameters collectively called $\theta$. Edges may also be featurized and influence the messages that are aggregated [24]. The node representations at the final propagation layer $K$ can be collected into the matrix $\boldsymbol{H} = \{\boldsymbol{h}_K(u) \mid u \in V\}$. Given a node $u \in G_q$ and a node $u' \in G_c$, we denote the embeddings of $u$ and $u'$ after the propagation layer $k$ as $\boldsymbol{h}_k^{(q)}(u)$ and $\boldsymbol{h}_k^{(c)}(u')$ respectively. $\boldsymbol{H}^{(q)}$ and $\boldsymbol{H}^{(c)}$ denote the $K$th-layer node embeddings of $G_q$ and $G_c$, collected into matrices. Note that, here the set of vectors $\boldsymbol{H}^{(q)}$ and $\boldsymbol{H}^{(c)}$ do not dependent on $G_c$ and $G_q$. In the end, IsoNet compares these embeddings to compute the distance $\Delta(G_c \mid G_q)$, which is inversely related to $\hat{y}(G_c \mid G_q)$.

$$\Delta(G_c \mid G_q) = \sum_{u,i} \text{ReLU}[\boldsymbol{H}^{(c)} - \boldsymbol{P}\boldsymbol{H}^{(q)}][u, i] \tag{2}$$

Since subgraph isomorphism entails an asymmetric relevance, we have: $\Delta(G_c \mid G_q) \neq \Delta(G_q \mid G_c)$. IsoNet also proposed another design of $\Delta$, where it replaces the node embeddings with edge embeddings and node alignment matrix with edge alignment matrix in Eq. (2).

In an **early** interaction network, $\boldsymbol{H}^{(q)}$ depends on $G_c$ and $\boldsymbol{H}^{(c)}$ depends on $G_q$ for any given $(G_q, G_c)$ pair. Formally, one should write $\boldsymbol{H}^{(q \mid c)}$ and $\boldsymbol{H}^{(c \mid q)}$ instead of $\boldsymbol{H}^{(q)}$ and $\boldsymbol{H}^{(c)}$ respectively for an early interaction network, but for simplicity, we will continue using $\boldsymbol{H}^{(q)}$ and $\boldsymbol{H}^{(c)}$.

**Our goal** Given a set of corpus graphs $C = \{G_c \mid c \in [|C|]\}$, our high-level goal is to build a graph retrieval model so that, given a query $G_q$, it can return the corpus graphs $\{G_c\}$ which are relevant to $G_q$. To that end, we seek to develop (1) a GNN-based early interaction model, and (2) an appropriate distance measure $\Delta(\cdot \mid \cdot)$, so that $\Delta(\boldsymbol{H}^{(c)} \mid \boldsymbol{H}^{(q)})$ is an accurate predictor of $y(G_c \mid G_q)$, at least to the extent that $\Delta(\cdot \mid \cdot)$ is effective for ranking candidate corpus graphs in response to a query graph.

## 3 Proposed early-interaction GNN with multi-round alignment refinement

In this section, we first write down the subgraph isomorphism task as an instance of the quadratic assignment problem (QAP) or the Gromov-Wasserstein (GW) distance optimization task. Then, we design IsoNet++, by building upon this formulation.

### 3.1 Subgraph isomorphism as Gromov-Wasserstein distance optimization

**QAP or GW formulation with asymmetric cost** We are given a graph pair $G_q$ and $G_c$ padded with appropriate number of nodes to ensure $|V_q| = |V_c| = n$ (say). Let their adjacency matrices be $\boldsymbol{A}_q, \boldsymbol{A}_c \in \{0, 1\}^{n \times n}$. Consider the family of hard permutation matrices $\boldsymbol{P} \in \Pi_n$ where $\boldsymbol{P}[u, u'] = 1$ indicates $u \in V_q$ is "matched" to $u' \in V_c$. Then, $G_q$ is a subgraph of $G_c$, if for some permutation matrix $\boldsymbol{P}$, the matrix $\boldsymbol{A}_q$ is covered by $\boldsymbol{P}\boldsymbol{A}_c\boldsymbol{P}^\top$, *i.e.*, for each pair $(u, v)$, whenever we have $\boldsymbol{A}_q[u, v] = 1$, we will also have $\boldsymbol{P}\boldsymbol{A}_c\boldsymbol{P}^\top[u, v] = 1$. This condition can be written as $\boldsymbol{A}_q \leq \boldsymbol{P}\boldsymbol{A}_c\boldsymbol{P}^\top$. We can regard a deficit in coverage as a cost or distance:

$$\text{cost}(\boldsymbol{P}; \boldsymbol{A}_q, \boldsymbol{A}_c) = \sum_{u \in [n], v \in [n]} \left[ \left( \boldsymbol{A}_q - \boldsymbol{P}\boldsymbol{A}_c\boldsymbol{P}^\top \right)_+ \right] [u, v] \tag{3}$$

$$= \sum_{u,v \in [n]} \sum_{u',v' \in [n]} (\boldsymbol{A}_q[u, v] - \boldsymbol{A}_c[u', v'])_+ \, \boldsymbol{P}[u, u'] \, \boldsymbol{P}[v, v'] \tag{4}$$

Here, $[\cdot]_+ = \max\{\cdot, 0\}$ is the ReLU function, applied elementwise. The function $\text{cost}(\boldsymbol{P}; \boldsymbol{A}_q, \boldsymbol{A}_c)$ can be driven down to zero using a suitable choice of $\boldsymbol{P}$ iff $G_q$ is a subgraph of $G_c$. This naturally suggests the relevance distance

$$\Delta(G_c \mid G_q) = \min_{\boldsymbol{P} \in \Pi_n} \text{cost}(\boldsymbol{P}; \boldsymbol{A}_q, \boldsymbol{A}_c) \tag{5}$$

Xu et al. [41] demonstrate that this QAP is a realization of the Gromov-Wassterstein distance minimization in a graph setting.

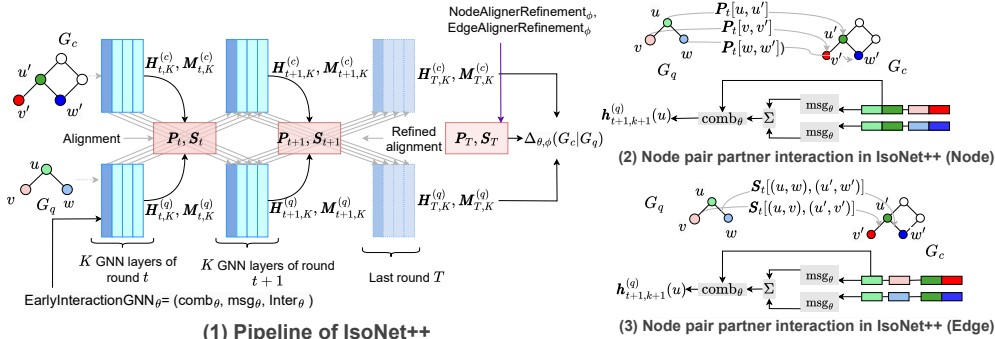

**(1) Pipeline of IsoNet++**

Figure 1: Overview of IsoNet++. Panel (a) shows the pipeline of IsoNet++. Given a graph pair $(G_q, G_c)$, we execute $T$ *rounds*, each consisting of $K$ GNN *layer* propagations. After a round $t$, we use the node embeddings to update the node alignment $\boldsymbol{P} = \boldsymbol{P}_t$ from its previous estimate $\boldsymbol{P} = \boldsymbol{P}_{t-1}$. Within each round $t \in [T]$, we compute the node embeddings of $G_q$ by gathering signals from $G_c$ and vice-versa, using GNN embeddings in the previous round and the node-alignment map $\boldsymbol{P}_t$. The alignment $\boldsymbol{P}_t$ remains consistent across all propagation layers $k \in [K]$ and is updated at the end of round $t$. Panel (b) shows our proposed node pair partner interaction in IsoNet++ (Node). When computing the message value of the node pair $(u, v)$, we also feed the node embeddings of the partners $u'$ and $v'$ in addition to the embeddings of the pairs $(u, v)$, where $u'$ and $v'$ is approximately aligned with $u$ and $v$, respectively (when converted to soft alignment, $u', v'$ need not be neighbors). Panel (c) shows the node pair partner interaction in IsoNet++ (Edge). In contrast to IsoNet++ (Node), here we feed the information from the message value of the partner pair $(u', v')$ instead of their node embeddings into the message passing network $\mathrm{msg}_\theta$.

**Updating $\boldsymbol{P}$ with projected gradient descent**  As shown in Benamou et al. [3], Peyré et al. [30], Xu et al. [41], one approach is to first relax $\boldsymbol{P}$ into a doubly stochastic matrix, which serves as a continuous approximation of the discrete permutation, and then update it using projected gradient descent (PGD). Here, the soft permutation $\boldsymbol{P}_{t-1}$ is updated to $\boldsymbol{P}_t$ at time-step $t$ by solving the following linear optimal transport (OT) problem, regularized with the entropy of $\{\boldsymbol{P}[u, v] \mid u, v \in [n]\}$ with a temperature $\tau$.

$$\boldsymbol{P}_t \leftarrow \underset{\boldsymbol{P} \in \mathcal{B}_n}{\arg\min} \, \mathrm{Trace}\left(\boldsymbol{P}^\top \nabla_{\boldsymbol{P}} \mathrm{cost}(\boldsymbol{P}; \boldsymbol{A}_q, \boldsymbol{A}_c)\big|_{\boldsymbol{P}=\boldsymbol{P}_{t-1}}\right) + \tau \sum_{u,v} \boldsymbol{P}[u, v] \cdot \log \boldsymbol{P}[u, v]. \quad (6)$$

Such an OT problem is solved using the iterative Sinkhorn-Knopp algorithm [10, 37, 26]. Similar to other combinatorial optimization problems on graphs, a QAP (4) does not capture the coverage cost in the presence of dense node or edge features, where two nodes or edges may exhibit graded degrees of similarity represented by continuous values. Furthermore, the binary values of the adjacency matrices result in inadequate gradient signals in $\nabla_{\boldsymbol{P}} \mathrm{cost}(\cdot)$. Additionally, the computational bottleneck of solving a fresh OT problem in each PGD step introduces a significant overhead, especially given the large number of pairwise evaluations required in typical learning-to-rank setups.

## 3.2  Design of IsoNet++ (Node)

Building upon the insights from the above GW minimization (3) and the successive refinement step (6), we build IsoNet++ (Node), the first variant of our proposed early interaction model.

**Node-pair partner interactions between graphs**  For simpler exposition, we begin by describing a synthetic scenario, where $\boldsymbol{P}$ is a hard node permutation matrix, which induces the alignment map as a bijection $\pi : V_q \rightarrow V_c$, so that $\pi(a) = b$ if $\boldsymbol{P}[a, b] = 1$. We first initialize layer $k = 0$ embeddings as $\boldsymbol{h}_0^{(q)}(u) = \mathrm{Init}_\theta(\mathrm{feature}(u))$ using a neural network $\mathrm{Init}_\theta$. (Throughout, $\boldsymbol{h}_k^{(c)}(u)$ are treated likewise.) Under the given alignment map $\pi$, a simple early interaction model would update the node embeddings as follows:

$$\boldsymbol{h}_{k+1}^{(q)}(u) = \mathrm{comb}_\theta\left(\boldsymbol{h}_k^{(q)}(u), \, \sum_{v \in \mathrm{nbr}(u)} \mathrm{msg}_\theta(\boldsymbol{h}_k^{(q)}(u), \boldsymbol{h}_k^{(q)}(v)), \, \boldsymbol{h}_k^{(c)}(\pi(u))\right) \quad (7)$$

In the above expression, the update layer uses representation of the partner node $u' \in V_c$ during the message passing step, to compute $\boldsymbol{h}_{k+1}^{(q)}(u)$, the embedding of node $u \in V_q$. Li et al. [22] use a similar update protocol, by approximating $\boldsymbol{h}_k^{(c)}(\pi(u)) = \sum_{u' \in V_c} a_{u' \rightarrow u}^{(k)} \boldsymbol{h}_k^{(c)}(u')$, where $a_{u' \rightarrow u}^{(k)}$ is the $k$th layer attention from $u \in V_q$ to potential partner $u' \in V_c$, with $\sum_{u' \in V_c} a_{u' \rightarrow u}^{(k)} = 1$. Instead

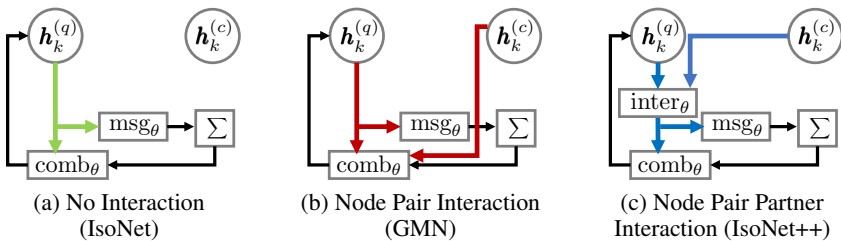

Figure 2: Illustration of the three interaction modes. IsoNet has no/late interaction between $\boldsymbol{h}^{(q)}$ and $\boldsymbol{h}^{(c)}$. IsoNet++ and GMN allow interaction between the representations of the query and corpus nodes. Under **node pair interaction**, the individual node embeddings $\boldsymbol{h}^{(q)}$ are used for message passing directly, thereby exposing them only to their neighbors. In the corresponding $\mathrm{comb}_\theta$ step, nodes interact only with their respective partners, therefore missing out on information from the partners of its neighbors. However, under **node pair partner interaction**, the representation of a node is combined with that of its partner(s) first, using the $\mathrm{inter}_\theta$ block to obtain $\boldsymbol{z}^{(q)}$ (12), which is used for message passing. Thus, when interacting with its neighbors, a node also gets information from the partners of its neighbors.

of regarding only nodes as potential partners, IsoNet++ will regard *node pairs* as partners. Given $(u, v) \in E_q$, the partners $(\pi(u), \pi(v)) \in E_c$ should then greatly influence the intensity of assimilation of $\boldsymbol{h}_k^{(c)}(u')$ into $\boldsymbol{h}_{k+1}^{(c)}(u)$. The first key innovation in IsoNet++ is to replace (7) to recognize and implement this insight:

$$\boldsymbol{h}_{k+1}^{(q)}(u) = \mathrm{comb}_\theta \left( [\boldsymbol{h}_k^{(q)}(u), \boldsymbol{h}_k^{(c)}(\pi(u))], \right.$$
$$\left. \sum_{v \in \mathrm{nbr}(u)} \mathrm{msg}_\theta \left( [\boldsymbol{h}_k^{(q)}(u), \boldsymbol{h}_k^{(c)}(\pi(u))], [\boldsymbol{h}_k^{(q)}(v), \boldsymbol{h}_k^{(c)}(\pi(v))] \right) \right) \quad (8)$$

Embeddings $\boldsymbol{h}_{k+1}^{(c)}(u')$ for nodes $u' \in V_c$ are updated likewise in a symmetric manner. The network $\mathrm{msg}_\theta$ is provided embeddings from partners $\pi(u), \pi(v)$ of $u, v \in V_q$ — this allows $\boldsymbol{h}_{k+1}^{(\bullet)}(u)$ to capture information from all nodes in the paired graph, that match with the $(k+1)$-hop neighbors of $u$. We schematically illustrate the interaction between the paired graphs in IsoNet, GMN and IsoNet++ in Figure 2.

**Multi-round lazy refinement of node alignment**    In reality, we are not given any alignment map $\pi$. This motivates our second key innovation beyond prior models [1, 22, 23, 35], where we decouple GNN layer propagation from updates to $\boldsymbol{P}$. To achieve this, IsoNet++ (Node) executes $T$ *rounds*, each consisting of $K$ *layer* propagations in both GNNs. At the end of each round $t$, we refine the earlier alignment $\boldsymbol{P}_{t-1}$ to the next estimate $\boldsymbol{P}_t$, which will be used in the next round. Henceforth, we will use the double subscript $t, k$ instead of the single subscript $k$ as in traditional GNNs. We denote the node embeddings at layer $k$ and round $t$ by $\boldsymbol{h}_{t,k}^{(q)}(u), \boldsymbol{h}_{t,k}^{(c)}(u') \in \mathbb{R}^{\dim_h}$ for $u \in V_q$ and $u' \in V_c$, which are (re-)initialized with node features $\boldsymbol{h}_{t,0}^{\bullet}$ for each round $t$. We gather these into matrices

$$\boldsymbol{H}_{t,k}^{(q)} = [\boldsymbol{h}_{t,k}^{(q)}(u) \mid u \in V_q] \in \mathbb{R}^{n \times \dim_h} \quad \text{and} \quad \boldsymbol{H}_{t,k}^{(c)} = [\boldsymbol{h}_{t,k}^{(c)}(u') \mid u' \in V_c] \in \mathbb{R}^{n \times \dim_h}. \quad (9)$$

$\boldsymbol{P}$ no longer remains an oracular hard permutation matrix, but becomes a doubly stochastic matrix indexed by rounds, written as $\boldsymbol{P}_t$. At the end of round $t$, a differentiable *aligner* module takes $\boldsymbol{H}_{t,K}^{(q)}$ and $\boldsymbol{H}_{t,K}^{(c)}$ as inputs and outputs a doubly stochastic node alignment (relaxed permutation) matrix $\boldsymbol{P}_t$ as follows:

$$\boldsymbol{P}_t = \mathrm{NodeAlignerRefinement}_\phi \left( \boldsymbol{H}_{t,K}^{(q)}, \boldsymbol{H}_{t,K}^{(c)} \right) \quad (10)$$

$$= \mathrm{GumbelSinkhorn} \left( \mathrm{LRL}_\phi(\boldsymbol{H}_{t,K}^{(q)}) \, \mathrm{LRL}_\phi(\boldsymbol{H}_{t,K}^{(c)})^\top \right) \in \mathcal{B}_n \quad (11)$$

In the above expression, $\mathrm{GumbelSinkhorn}(\bullet)$ performs iterative Sinkhorn normalization on the input matrix added with Gumbel noise [26]; $\mathrm{LRL}_\phi$ is a neural module consisting of two linear layers with a ReLU activation after the first layer. As we shall see next, $\boldsymbol{P}_t$ is used to gate messages flowing *across* from one graph to the other during round $t+1$, i.e., while computing $\boldsymbol{H}_{t+1,1:K}^{(q)}$ and $\boldsymbol{H}_{t+1,1:K}^{(c)}$. The soft alignment $\boldsymbol{P}_t$ is kept frozen for the duration of all layers in round $t+1$. $\boldsymbol{P}_t[u, u']$ may be interpreted as the probability that $u$ is assigned to $u'$, which naturally requires that $\boldsymbol{P}_t$ should be

row-equivariant (column equivariant) to the shuffling of the node indices of $G_q$ ($G_c$). As shown in Appendix D, the above design choice (11) ensures this property.

**Updating node representation using early-interaction GNN** Here, we describe the early interaction GNN for the query graph $G_q$. The GNN on the corpus graph $G_c$ follows the exact same design and is deferred to Appendix E.1. In the initial round ($t = 1$), since there is no prior alignment estimate $\boldsymbol{P}_{t=0}$, we employ the traditional late interaction GNN (1) to compute all layers $\boldsymbol{H}_{1,1:K}^{(q)}$ and $\boldsymbol{H}_{1,1:K}^{(c)}$ separately. These embeddings are then used to estimate $\boldsymbol{P}_{t=1}$ using Eq. (11). For subsequent rounds ($t > 1$), given embeddings $\boldsymbol{H}_{t,1:K}^{(q)}$, and the alignment estimate matrix $\boldsymbol{P}_t$, we run an early interaction GNN from scratch. We start with a fresh initialization of the node embeddings as before; i.e., $\boldsymbol{h}_{t+1,0}^{(q)}(u) = \text{Init}_\theta(\text{feature}(u))$. For each subsequent propagation layer $k + 1$ ($k \in [0, K - 1]$), we approximate (8) as follows. We read previous-round, same-layer embeddings $\boldsymbol{h}_{t,k}^{(c)}(u')$ of nodes $u'$ from the other graph $G_c$, incorporate the alignment strength $\boldsymbol{P}_t[u, u']$, and aggregate these to get an intermediate representation of $u$ that is sensitive to $\boldsymbol{P}_t$ and $G_c$.

$$\boldsymbol{z}_{t+1,k}^{(q)}(u) = \text{inter}_\theta \left( \boldsymbol{h}_{t+1,k}^{(q)}(u), \sum_{u' \in V_c} \boldsymbol{h}_{t,k}^{(c)}(u') \boldsymbol{P}_t[u, u'] \right) \qquad (12)$$

Here, $\text{inter}_\theta$ is a neural network that computes interaction between the graph pairs; $\boldsymbol{z}_{t+1,k}^{(q)}(u)$ provides a soft alignment guided representation of $[\boldsymbol{h}_k^{(q)}(u), \boldsymbol{h}_k^{(c)}(\pi(u))]$ in Eq. (8), which can be relaxed as:

$$\boldsymbol{h}_{t+1,k+1}^{(q)}(u) = \text{comb}_\theta \left( \boldsymbol{z}_{t+1,k}^{(q)}(u), \sum_{v \in \text{nbr}(u)} \text{msg}_\theta(\boldsymbol{z}_{t+1,k}^{(q)}(u), \boldsymbol{z}_{t+1,k}^{(q)}(v)) \right) \qquad (13)$$

In the above expression, we explicitly feed $\boldsymbol{z}_{t+1,k}^{(q)}(v), v \in \text{nbr}(u)$ in the $\text{msg}_\theta$ network, capturing embeddings of nodes in the corpus $G_c$ aligned with the *neighbors* of node $u \in V_q$ in $\boldsymbol{h}_{t+1,k+1}^{(q)}(u)$. This allows the model to perform node-pair partner interaction. Instead, if we were to feed only $\boldsymbol{h}_{t+1,k}^{(q)}(u)$ into the $\text{msg}_\theta$ network, then it would only perform node partner interaction. In this case, the computed embedding for $u$ would be based solely on signals from nodes in the paired graph that directly correspond to $u$, therefore missing additional context from other neighbourhood nodes.

**Distant supervision of alignment** Finally, at the end of $T$ rounds, we express the relevance distance $\Delta(G_c \,|\, G_q)$ as a soft distance between the set $\boldsymbol{H}_{T,K}^{(q)} = [\boldsymbol{h}_{T,K}^{(q)}(u) \,|\, u \in V_q]$ and $\boldsymbol{H}_{T,K}^{(c)} = [\boldsymbol{h}_{T,K}^{(c)}(u') \,|\, u' \in V_c]$, measured as

$$\Delta_{\theta,\phi}(G_c \,|\, G_q) = \sum_u \sum_d \text{ReLU}(\boldsymbol{H}_{T,K}^{(q)}[u, d] - (\boldsymbol{P}_T \boldsymbol{H}_{T,K}^{(c)})[u, d]) \qquad (14)$$

Our focus is on graph retrieval applications. It is unrealistic to assume direct supervision from a gold alignment map $\boldsymbol{P}^*$. Instead, training query instances are associated with pairwise preferences between two corpus graphs, in the form $\langle G_q, G_{c+}, G_{c-} \rangle$, meaning that, ideally, we want $\Delta_{\theta,\phi}(G_{c-}|G_q) \geq \gamma + \Delta_{\theta,\phi}(G_{c+}|G_q)$, where $\gamma > 0$ is a margin hyperparameter. This suggests a minimization of the standard hinge loss as follows:

$$\min_{\theta,\phi} \sum_{q \in Q} \sum_{c+ \in C_{q+}, c- \in C_{q-}} [\gamma + \Delta_{\theta,\phi}(G_{c+} \,|\, G_q) - \Delta_{\theta,\phi}(G_{c-} \,|\, G_q)]_+ \qquad (15)$$

This loss is back-propagated to train model weights $\theta$ in $\text{comb}_\theta, \text{inter}_\theta, \text{msg}_\theta$ and weights $\phi$ in the Gumbel-Sinkhorn network.

**Multi-layer eager alignment variant** Having set up the general multi-round framework of IsoNet++, we introduce a structurally simpler variant that updates $\boldsymbol{P}$ eagerly after every layer, eliminating the need to re-initialize node embeddings every time we update $\boldsymbol{P}$. The eager variant retains the benefits of node-pair partner interactions, while ablating IsoNet++ toward GMN. Updating $\boldsymbol{P}$ via Sinkhorn iterations is expensive compared to a single GNN layer. In practice, we see a non-trivial tradeoff between computation cost, end task accuracy, and the quality of our injective alignments, depending on the value of $K$ for eager updates, and the values $(T, K)$ for lazy updates. Formally, $\boldsymbol{P}_k$ is updated across layers as follows:

$$\boldsymbol{P}_k = \text{NodeAlignerRefinement}_\phi \left( \boldsymbol{H}_k^{(q)}, \boldsymbol{H}_k^{(c)} \right) \qquad (16)$$

$$= \text{GumbelSinkhorn} \left( \text{LRL}_\phi(\boldsymbol{H}_k^{(q)}) \, \text{LRL}_\phi(\boldsymbol{H}_k^{(c)})^\top \right). \qquad (17)$$

We update the GNN embeddings, layerwise, as follows:

$$z_k^{(q)}(u) = \text{inter}_\theta \left( h_k^{(q)}(u), \sum_{u' \in V_c} h_k^{(c)}(u') P_k[u, u'] \right), \tag{18}$$

$$h_{k+1}^{(q)}(u) = \text{comb}_\theta \left( z_k^{(q)}(u), \sum_{v \in \text{nbr}(u)} \text{msg}_\theta(z_k^{(q)}(u), z_k^{(q)}(v)) \right) \tag{19}$$

**Analysis of computational complexity**  Here, will compare the performance of IsoNet (Node) [35] with multi-layer IsoNet++ (Node) and multi-round IsoNet++ (Node) for graphs with $|V|$ nodes. For multi-layer IsoNet++ (Node) and IsoNet (Node), we assume $K$ propagation steps and for multi-round IsoNet++ (Node), $T$ rounds, each with $K$ propagation steps.

—*IsoNet (Node):* The total complexity is $O(|V|^2 + K|E|)$, computed as follows: **(1)** Initialization of layer embeddings at layer $k = 0$ takes $O(|V|)$ time. **(2)** The node representation computation incurs a complexity of $O(|E|)$ for each message passing step since it aggregates node embeddings across all neighbors. **(3)** The computation of $P$ takes $O(|V|^2)$ time.

—*Multi-layer eager IsoNet++ (Node):* The total complexity is $O(K|V|^2 + K|E| + K|V|^2) = O(K|V|^2)$, computed as follows: **(1)** Initialization (layer $k = 0$) takes $O(|V|)$ time. **(2)** The computation of intermediate embeddings $z^{(\bullet)}$ (Eq. 18) involves the evaluation of the expression $\sum_{u' \in V_c} h_k^{(\bullet)}(u') P_k[u, u']$ and hence admits a complexity of $O(|V|)$ for each node per layer. The total complexity for $K$ steps and $|V|$ nodes is thus $O(K|V|^2)$. **(3)** Next, for each node in every layer, we compute $h_{k+1}^{(\bullet)}$ (Eq. 19) which gathers messages $z^{(\bullet)}$ from all its neighbors, contributing a total complexity of $O(K|E|)$. **(4)** Finally, we update $P_k$ for each layer which has a complexity of $O(K|V|^2)$.

—*Multi-round IsoNet++ (Node):* Here, the key difference from the multi-layer version above is that the doubly stochastic matrix $P_t$ from round $t$ is used to compute $z$ and the $K$-step-GNN runs in each of the $T$ rounds. This multiplies the complexity of steps 2 and 3 with $T$, raising it to $O(KT|V|^2 + KT|E|)$. Matrix $P_t$ is updated a total of $T$ times, which changes the complexity of step 4 to $O(T|V|^2)$. Hence, the total complexity is $O(KT|V|^2 + T|V|^2 + KT|E|) = O(KT|V|^2)$.

Hence, the complexity of IsoNet is $O(|V|^2 + K|E|)$, multi-layer IsoNet++ is $O(K|V|^2)$ and multi-round IsoNet++ is $O(KT|V|^2)$. This increased complexity of the latter comes with the benefit of a significant performance boost, as our experiments suggest.

### 3.3  Extension of IsoNet++ (Node) to IsoNet++ (Edge)

We now extend IsoNet++ (Node) to IsoNet++ (Edge) which uses explicit edge alignment for interaction across GNN and relevance distance surrogate.

**Multi-round refinement of edge alignment**  In IsoNet++ (Edge), we maintain a soft edge permutation matrix $S$ which is frozen at $S = S_{t-1}$ within each round $t \in [T]$ and gets refined after every round $t$ as $S_{t-1} \to S_t$. Similar to IsoNet++ (Node), within each round $t$, GNN runs from scratch: it propagates messages across layers $k \in [K]$ and $S_{t-1}$ assists it to capture cross-graph signals. Here, in addition to node embeddings $h_{t,k}^{(\bullet)}$, we also use edge embeddings $m_{t,k}^{(q)}(e), m_{t,k}^{(c)}(e') \in \mathbb{R}^{\dim m}$ at each layer $k$ and each round $t$, which capture the information about the subgraph $k \le K$ hop away from the edges $e$ and $e'$. Similar to Eq. (9), we define $M_{t,k}^{(q)} = [m_{t,k}^{(q)}(e)]_{e \in E_q}$, and $M_{t,k}^{(c)} = [m_{t,k}^{(c)}(e')]_{e' \in E_c}$. $M_{t,0}^{(\bullet)}$ are initialized using the features of the nodes connected by the edges, and possibly local edge features. Given the embeddings $M_{t,K}^{(q)}$ and $M_{t,K}^{(c)}$ computed at the end of round $t$, an edge aligner module ($\text{EdgeAlignerRefinement}_\phi(\bullet)$) takes these embedding matrices as input and outputs a soft edge permutation matrix $S_t$, similar to the update of $P_t$ in Eq. (11).

$$S_t = \text{EdgeAlignerRefinement}_\phi \left( M_{t,K}^{(q)}, M_{t,K}^{(c)} \right) \tag{20}$$

$$= \text{GumbelSinkhorn}(\text{LRL}_\phi(M_{t,K}^{(q)}) \, \text{LRL}_\phi(M_{t,K}^{(c)})^\top) \tag{21}$$

Here, $M_{t,K}^{(\bullet)}$ are appropriately padded to ensure that they have the same number of rows.

**Edge alignment-induced early interaction GNN**  For $t = 1$, we start with a late interaction model using vanilla GNN (1) and obtain $S_{t=1}$ using Eq. (21). Having computed the edge embeddings $m_{t,1:K}^{(\bullet)}(\bullet)$ and node embeddings $h_{t,1:K}^{(\bullet)}(\bullet)$ upto round $t$, we compute $S_t$ and use it to build a fresh early interaction GNN for round $t + 1$. To this end, we adapt the GNN guided by $P_t$ in Eqs. (12)–

(13),to the GNN guided by $\boldsymbol{S}_t$. We overload the notations for neural modules and different embedding vectors from IsoNet++ (Node), whenever their roles are similar.

Starting with the same initialization as in IsoNet++ (Node), we perform the cross-graph interaction guided by the soft edge permutation matrix $\boldsymbol{S}_t$, similar to Eq. (12). Specifically, we use the embeddings of edges $\{e' = (u', v')\} \in E_c$, computed at layer $k$ at round $t$, which share soft alignments with an edge $e = (u, v) \in E_q$, to compute $\boldsymbol{z}_{t+1,k}^{(q)}(e)$ and $\boldsymbol{z}_{t+1,k}^{(q)}(e')$ as follows:

$$\boldsymbol{z}_{t+1,k}^{(q)}(e) = \text{inter}_\theta \left( \boldsymbol{m}_{t+1,k}^{(q)}(e), \sum_{e' \in E_c} \boldsymbol{m}_{t,k}^{(c)}(e') \boldsymbol{S}_t[e, e'] \right) \tag{22}$$

Finally, we update the node embeddings $\boldsymbol{h}_{t+1,k+1}^{(\bullet)}$ for propagation layer $k+1$ as

$$\boldsymbol{h}_{t+1,k+1}^{(q)}(u) = \text{comb}_\theta \left( \boldsymbol{h}_{t+1,k}^{(q)}(u), \sum_{a \in \text{nbr}(u)} \text{msg}_\theta(\boldsymbol{h}_{t+1,k}^{(q)}(u), \boldsymbol{h}_{t+1,k}^{(q)}(a), \boldsymbol{z}_{t+1,k}^{(q)}((u, a))) \right) \tag{23}$$

In this case, we perform the cross-graph interaction at the edge level rather than the node level. Hence, $\text{msg}_\theta$ acquires cross-edge signals separately as $\boldsymbol{z}_{t+1,k}^{(\bullet)}$. Finally, we use $\boldsymbol{h}_{t+1,k+1}^{(\bullet)}$ and $\boldsymbol{z}_{t+1,k+1}^{(\bullet)}$ to update $\boldsymbol{m}_{t+1,k+1}^{(\bullet)}$ as follows:

$$\boldsymbol{m}_{t+1,k+1}^{(q)}((u, v)) = \text{msg}_\theta \left( \boldsymbol{h}_{t+1,k+1}^{(q)}(u), \boldsymbol{h}_{t+1,k+1}^{(q)}(v), \boldsymbol{z}_{t+1,k}^{(q)}((u, v)) \right) \tag{24}$$

Likewise, we develop $\boldsymbol{m}_{t+1,k+1}^{(c)}$ for corpus graph $G_c$. Note that $\boldsymbol{m}_{t+1,k+1}^{(q)}((u, v))$ captures signals not only from the matched pair $(u', v')$, but also signals from the nodes in $G_c$ which share correspondences with the neighbor nodes of $u$ and $v$. Finally, we pad zero vectors to $[\boldsymbol{m}_{T,K}^{(q)}(e)]_{e \in E_q}$ and $[\boldsymbol{m}_{T,K}^{(c)}(e')]_{e' \in E_c}$ to build the matrices $\boldsymbol{M}_{T,K}^{(q)}$ and $\boldsymbol{M}_{T,K}^{(c)}$ with same number of rows, which are finally used to compute the relevance distance

$$\Delta_{\theta,\phi}(G_c \mid G_q) = \sum_u \sum_d \text{ReLU}(\boldsymbol{M}_{T,K}^{(q)}[e, d] - (\boldsymbol{S}_T \boldsymbol{M}_{T,K}^{(c)})[e, d]). \tag{25}$$

## 4 Experiments

We report on a comprehensive evaluation of IsoNet++ on six real datasets and analyze the efficacy of the key novel design choices. In Appendix G, we provide results of additional experiments.

### 4.1 Experimental setup

**Datasets** We use six real world datasets in our experiments, *viz.*, AIDS, Mutag, PTC-FM (FM), PTC-FR (FR), PTC-MM (MM) and PTC-MR (MR), which were also used in [27, 35]. Appendix F provides the details about dataset generation and their statistics.

**State-of-the-art baselines** We compare our method against eleven state-of-the-art methods, *viz.*, (1) GraphSim [2] (2) GOTSim [11], (3) SimGNN [1], (4) EGSC [31], (5) H2MN [45], (6) Neuromatch [23], (7) GREED [32], (8) GEN [22], (9) GMN [22] (10) IsoNet (Node) [35], and (11) IsoNet (Edge) [35]. Among them, Neuromatch, GREED, IsoNet (Node) and IsoNet (Edge) apply asymmetric hinge distances between query and corpus embeddings for $\Delta(G_c \mid G_q)$, specifically catered towards subgraph matching, similar to our method in Eqs. (14) and (25). GMN and GEN use symmetric Euclidean distance between their (whole-) graph embeddings $\boldsymbol{g}^{(q)}$ (for query) and $\boldsymbol{g}^{(c)}$ (for corpus) as $||\boldsymbol{g}^{(q)} - \boldsymbol{g}^{(c)}||$ in their paper [22], which is not suitable for subgraph matching and therefore, results in poor performance. Hence, we change it to $\Delta(G_c \mid G_q) = [\boldsymbol{g}^{(q)} - \boldsymbol{g}^{(c)}]_+$. The other methods first compute the graph embeddings, then fuse them using a neural network and finally apply a nonlinear function on the fused embeddings to obtain the relevance score.

**Training and evaluation protocol** Given a fixed corpus set $C$, we split the query set $Q$ into $60\%$ training, $15\%$ validation and $25\%$ test set. We train all the models on the training set by minimizing a ranking loss (15). During the training of each model, we use five random seeds. Given a test query $q'$, we rank the corpus graphs $C$ in the decreasing order of $\Delta_{\theta,\phi}(G_c \mid G_{q'})$ computed using the trained model. We evaluate the quality of the ranking by measuring Average Precision (AP) and HITS@20, described in Appendix F. Finally, we report mean average precision (MAP) and mean HITS@20, across all the test queries. By default, we set the number of rounds $T = 3$, the number of propagation layers in GNN $K = 5$. In Appendix F, we discuss the baselines, hyperparameter setup and the evaluation metrics in more detail.

| Metrics → | Mean Average Precision (MAP) | | | | | | HITS @ 20 | | | | | |
|---|---|---|---|---|---|---|---|---|---|---|---|---|
| | AIDS | Mutag | FM | FR | MM | MR | AIDS | Mutag | FM | FR | MM | MR |
| GraphSim [2] | 0.356 | 0.472 | 0.477 | 0.423 | 0.415 | 0.453 | 0.145 | 0.257 | 0.261 | 0.227 | 0.212 | 0.23 |
| GOTSim [11] | 0.324 | 0.272 | 0.355 | 0.373 | 0.323 | 0.317 | 0.112 | 0.088 | 0.147 | 0.166 | 0.119 | 0.116 |
| SimGNN [1] | 0.341 | 0.283 | 0.473 | 0.341 | 0.298 | 0.379 | 0.138 | 0.087 | 0.235 | 0.155 | 0.111 | 0.160 |
| EGSC [31] | 0.505 | 0.476 | 0.609 | 0.607 | 0.586 | 0.58 | 0.267 | 0.243 | 0.364 | 0.382 | 0.348 | 0.325 |
| H2MN [45] | 0.267 | 0.276 | 0.436 | 0.412 | 0.312 | 0.243 | 0.076 | 0.084 | 0.200 | 0.189 | 0.119 | 0.069 |
| Neuromatch [23] | 0.489 | 0.576 | 0.615 | 0.559 | 0.519 | 0.606 | 0.262 | 0.376 | 0.389 | 0.350 | 0.282 | 0.385 |
| GREED [32] | 0.472 | 0.567 | 0.558 | 0.512 | 0.546 | 0.528 | 0.245 | 0.371 | 0.316 | 0.287 | 0.311 | 0.277 |
| GEN [22] | 0.557 | 0.605 | 0.661 | 0.575 | 0.539 | 0.631 | 0.321 | 0.429 | 0.448 | 0.368 | 0.292 | 0.391 |
| GMN [22] | 0.622 | 0.710 | 0.730 | 0.662 | 0.655 | 0.708 | 0.397 | 0.544 | 0.537 | 0.45 | 0.423 | 0.49 |
| IsoNet (Node) [35] | 0.659 | 0.697 | 0.729 | 0.68 | 0.708 | 0.738 | 0.438 | 0.509 | 0.525 | 0.475 | 0.493 | 0.532 |
| IsoNet (Edge) [35] | 0.690 | 0.706 | 0.783 | 0.722 | 0.753 | 0.774 | 0.479 | 0.529 | 0.613 | 0.538 | 0.571 | 0.601 |
| IsoNet++ (Node) | 0.825 | 0.851 | 0.888 | 0.855 | 0.838 | 0.874 | 0.672 | 0.732 | 0.797 | 0.737 | 0.702 | 0.755 |
| IsoNet++ (Edge) | 0.847 | 0.858 | 0.902 | 0.875 | 0.902 | 0.902 | 0.705 | 0.749 | 0.813 | 0.769 | 0.809 | 0.803 |

Table 3: Comparison of the two variants of IsoNet++ (IsoNet++ (Node) and IsoNet++ (Edge)) against all the state-of-the-art graph retrieval methods, across all six datasets. Performance is measured in terms average precision (MAP) and mean HITS@20. In all cases, we used 60% training, 15% validation and 25% test sets. The numbers highlighted with green and yellow indicate the best, second best method respectively, whereas the numbers with blue indicate the best method among the baselines. (MAP values for IsoNet++ (Edge) across FM, MM and MR were verified to be not exactly the same, but they match up to the third decimal place.)

| | | AIDS | Mutag | FM | FR | MM | MR |
|---|---|---|---|---|---|---|---|
| Node | Eager | 0.756 | 0.81 | 0.859 | 0.802 | 0.827 | 0.841 |
| | Lazy | 0.825 | 0.851 | 0.888 | 0.855 | 0.838 | 0.874 |
| Edge | Eager | 0.795 | 0.805 | 0.883 | 0.812 | 0.862 | 0.886 |
| | Lazy | 0.847 | 0.858 | 0.902 | 0.875 | 0.902 | 0.902 |

Table 4: Lazy multi-round vs. eager multi-layer. First (Last) two rows report MAP for IsoNet++ (Node) (IsoNet++ (Edge)). Green shows the best method.

| | | AIDS | Mutag | FM | FR | MM | MR |
|---|---|---|---|---|---|---|---|
| Lazy | Node partner | 0.776 | 0.829 | 0.851 | 0.819 | 0.844 | 0.84 |
| | IsoNet++ (Node) | 0.825 | 0.851 | 0.888 | 0.855 | 0.838 | 0.874 |
| Eager | Node partner | 0.668 | 0.783 | 0.821 | 0.752 | 0.753 | 0.794 |
| | IsoNet++ (Node) | 0.756 | 0.81 | 0.859 | 0.802 | 0.827 | 0.841 |

Table 5: Node partner vs. node pair partner interaction. First (Last) two rows report MAP for multi-round (multi-layer) update. Green shows the best method.

## 4.2 Results

**Comparison with baselines** First, we compare IsoNet++ (Node) and IsoNet++ (Edge) against all the baselines, across all datasets. In Table 3, we report the results. The key observations are as follows: **(1)** IsoNet++ (Node) and IsoNet++ (Edge) outperform all the baselines by significant margins across all datasets. IsoNet++ (Edge) consistently outperforms IsoNet++ (Node). This is because edge alignment allows us to compare the graph pairs more effectively than node alignment. A similar effect was seen for IsoNet (Edge) vs. IsoNet (Node) [35]. **(2)** Among all state-of-the-art competitors, IsoNet (Edge) performs the best followed by IsoNet (Node). Similar to us, they also use edge and node alignments respectively. However, IsoNet does not perform any interaction between the graph pairs and the alignment is computed once only during the computation of $\Delta(G_c \mid G_q)$. This results in modest performance compared to IsoNet++. **(3)** GMN uses "attention" to estimate the alignment between graph pairs, which induces a non-injective mapping. Therefore, despite being an early interaction model, it is mostly outperformed by IsoNet, which uses injective alignments.

**Lazy vs. eager updates** In lazy multi-round updates, the alignment matrices remain unchanged across all propagation layers and are updated only after the GNN completes its $K$-layer message propagations. To evaluate its effectiveness, we compare it against the eager multi-*layer* update (described at the end of Section 3.2), where the GNN executes its $K$-layer message propagations only once; the alignment map is updated across $K$ layers; and, the alignment at $k$th layer is used to compute the embeddings at $(k+1)$th layer. In Table 4, we compare the performance in terms MAP, which shows that lazy multi-round updates significantly outperform multi-layer updates.

**Node partner vs. node-pair partner interaction** To understand the benefits of node-pair partner interaction, we contrast IsoNet++ (Node) against another variant of our method, which performs *node partner* interaction rather than node pair partner interaction, similar to Eq. (7). For lazy multi-round updates, we compute the embeddings as follows:

$$\boldsymbol{h}_{t+1,k+1}^{(q)}(u) = \text{comb}_\theta(\boldsymbol{h}_{t+1,k}^{(q)}(u), \ \sum_{v \in \text{nbr}(u)} \text{msg}_\theta(\boldsymbol{h}_{t,k}^{(q)}(u), \boldsymbol{h}_{t,k}^{(q)}(v)), \ \sum_{u' \in V_c} \boldsymbol{P}_t[u,u']\boldsymbol{h}_{t,k}^{(c)}(u'))$$

For eager multi-layer updates, we compute the embeddings as:

$$\boldsymbol{h}_{k+1}^{(q)}(u) = \text{comb}_\theta(\boldsymbol{h}_k^{(q)}(u), \ \sum_{v \in \text{nbr}(u)} \text{msg}_\theta(\boldsymbol{h}_k^{(q)}(u), \boldsymbol{h}_k^{(q)}(v)), \ \sum_{u' \in V_c} \boldsymbol{P}_k[u,u']\boldsymbol{h}_k^{(c)}(u'))$$

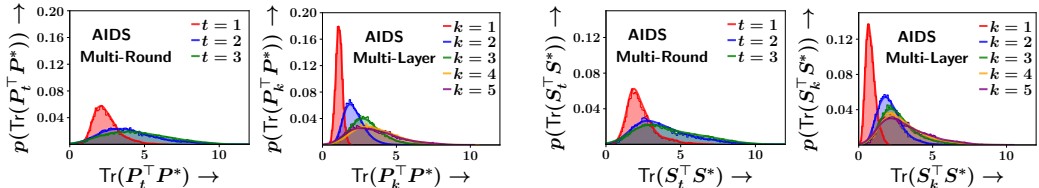

Figure 6: Empirical probability density of similarity between the estimated alignments and the true alignments $\boldsymbol{P}^*, \boldsymbol{S}^*$ for both multi-round and multi-layer update strategies across different stages of updates ($t$ for multi-round and $k$ for multi-layer), for AIDS. Similarity is measured using $p(\mathrm{Tr}(\boldsymbol{P}_t^\top \boldsymbol{P}^*)), p(\mathrm{Tr}(\boldsymbol{S}_t^\top \boldsymbol{S}^*))$ for multi-round lazy updates and $p(\mathrm{Tr}(\boldsymbol{P}_k^\top \boldsymbol{P}^*)), p(\mathrm{Tr}(\boldsymbol{S}_k^\top \boldsymbol{S}^*))$ for multi-layer eager updates.

Table 5 summarizes the results, which shows that IsoNet++ (Node) (node partner pair) performs significantly better than Node partner for both multi-round lazy updates (top-two rows) and multi-layer eager updates (bottom tow rows).

**Quality of injective alignments**  Next we compare between multi-round and multi-layer update strategies in terms of their ability to refine the alignment matrices, as the number of updates of these matrices increases. For multi-round (layer) updates, we instrument the alignments $\boldsymbol{P}_t$ and $\boldsymbol{S}_t$ ($\boldsymbol{P}_k$ and $\boldsymbol{S}_k$) for different rounds $t \in [T]$ (layers $k \in [K]$). Specifically, we look into the distribution of the similarity between the learned alignments $\boldsymbol{P}_t, \boldsymbol{S}_t$ and the correct alignments $\boldsymbol{P}^*, \boldsymbol{S}^*$ (using combinatorial routine), measured using the inner products $\mathrm{Tr}(\boldsymbol{P}_t^\top \boldsymbol{P}^*)$ and $\mathrm{Tr}(\boldsymbol{S}_t^\top \boldsymbol{S}^*)$ for different $t$. Similarly, we compute $\mathrm{Tr}(\boldsymbol{P}_k^\top \boldsymbol{P}^*)$ and $\mathrm{Tr}(\boldsymbol{S}_k^\top \boldsymbol{S}^*)$ for different $k \in [K]$. Figure 6 summarizes the results, which shows that **(1)** as $t$ or $k$ increases, the learned alignments become closer to the gold alignments; **(2)** multi-round updates refine the alignments approximately twice as faster than the multi-layer variant. The distribution of $\mathrm{Tr}(\boldsymbol{P}_t^\top \boldsymbol{P}^*)$ at $t = 1$ in multi-round strategy is almost always close to $\mathrm{Tr}(\boldsymbol{P}_k^\top \boldsymbol{P}^*)$ for $k = 2$. Note that, our aligner networks learn to refine the $\boldsymbol{P}_t$ and $\boldsymbol{S}_t$ through end-to-end training, without using any form of supervision from true alignments or the gradient computed in Eq. (6).

**Accuracy-inference time trade-off**  Here, we analyze the accuracy and inference time trade-off. We vary $T$ and $K$ for IsoNet++'s lazy multi-round variant, and vary $K$ for IsoNet++'s eager multi-layer variant and for GMN. Figure 7 summarizes the results. Notably, the eager multi-layer variant achieves the highest accuracy for $K = 8$ on the AIDS dataset, despite the known issue of oversmoothing in GNNs for large $K$. This unexpected result may be due to our message passing components, which involve terms like $\sum_{u'} \boldsymbol{P}[u, u']\boldsymbol{h}(u')$, effectively acting as a convolution between alignment scores and embedding vectors. This likely enables $\boldsymbol{P}$ to function as a filter, countering the oversmoothing effect.

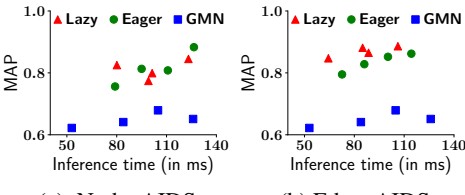

(a) Node, AIDS  (b) Edge, AIDS

Figure 7: Trade-off between MAP and inference time (batch size=128).

## 5   Conclusion

We introduce IsoNet++ as an early-interaction network for estimating subgraph isomorphism. IsoNet++ learns to identify explicit alignments between query and corpus graphs despite having access to only pairwise preferences and not explicit alignments during training. We design a graph neural network (GNN) that uses an alignment estimate to propagate messages, then uses the GNN's output representations to refine the alignment. Experiments across several datasets confirm that alignment refinement is achieved over several rounds. Design choices such as using node-pair partner interaction (instead of node partner) and lazy updates (over eager) boost the performance of our architecture, making it the state-of-the-art in subgraph isomorphism based subgraph retrieval. We also demonstrate the accuracy v/s inference time trade offs for IsoNet++, which show how different knobs can be tuned to utilize our models under regimes with varied time constraints.

This study can be extended to graph retrieval problems which use different graph similarity measures, such as maximum common subgraph or graph edit distance. Extracting information from node-pairs is effective and can be widely used to improve GNNs working on multiple graphs at once.

## Acknowledgements

Indradyumna acknowledges Qualcomm Innovation Fellowship, Abir and Soumen acknowledge grants from Amazon, Google, IBM and SERB.

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

# Iteratively Refined Early Interaction Alignment for Subgraph Matching based Graph Retrieval
## (Appendix)

## A  Limitations

We find two limitations of our method each of which could form the basis of detailed future studies.

1. Retrieval systems greatly benefit from the similarity function being hashable. This can improve the inference time multi-fold while losing very little, if at all any, performance, making the approach ready for production environments working under tight time constraints. The design of a hash function for an early interaction network like ours is unknown and seemingly difficult. In fact, such a hashing procedure is not known even for predecessors like IsoNet (Edge) or GMN, and this is an exciting future direction.
2. Our approach does not explicitly differentiate between nodes or edges that may belong to different classes. This can be counterproductive when there exist constraints that prevent the alignment of two nodes or edges with different labels. While the network is designed to process node and edge features, it might not be enough to rule out alignments that violate the said constraint. Such constraints could also exist for node-pairs, such as in knowledge graphs with hierarchical relationships between entity types, and are not taken into account by our model. Extending our work to handle such restrictions is an interesting problem to consider.

## B  Related work

In this section, we discuss different streams of work that are related to and have influenced the study.

### B.1  Graph Representation Learning

Graph neural networks (GNN) [14, 22, 21, 18, 42, 38] have emerged as a widely applicable approach for graph representation learning. A graph neural network computes the embedding of a node by aggregating the representations of its neighbors across $K$ steps of message passing, effectively combining information from $K$-hop neighbors. GNNs were first used for graph similarity computation by Li et al. [22], who enriched the architecture with attention to predict isomorphism between two graphs. Attention acts as a mechanism to transfer information from the representation of one graph to that of the other, thus boosting the performance of the approach. Chen et al. [7] enriched the representation of graphs by capturing the subgraph around a node effectively through a structure aware transformer architecture.

### B.2  Differentiable combinatorial solvers

We utilize a differentiable gadget to compute an injective alignment, which is a doubly stochastic matrix. The differentiability is crucial to the training procedure as it enables us to backpropagate through the alignments. The GumbelSinkhorn operator, which performs alternating normalizations across rows and columns, was first proposed by Sinkhorn and Knopp [37] and later used for the Optimal Transport problem by Cuturi [10]. Other methods to achieve differentiability include adding random noise to the inputs to discrete solvers [4] and designing probabilistic loss functions [17]. A compilation of such approaches towards constrained optimization on graphs through neural techniques is presented in [19].

### B.3  Graph Similarity Computation and Retrieval

Several different underlying measures have been proposed for graph similarity computation, including full graph isomorphism [22], subgraph isomorphism [23, 35], graph edit distance (GED) [2, 11, 13, 28, 44] and maximum common subgraph (MCS) [2, 11, 34]. Bai et al. [2] proposed GraphSim towards the GED and MCS problems, using convolutional neural network based scoring on top of graph similarity matrices. GOTSim [11] explicitly computes the alignment between the two graphs by studying the optimal transformation cost. GraphSim [2] utilizes both graph-level and node-level signals to compute a graph similarity score. NeuroMatch [23] evaluates, for each node pair across the two graphs, if the neighborhood of one node is contained in the neighborhood of another using order embeddings [25]. GREED [32] proposed a Siamese graph isomorphism network, a late interaction

model to tackle the GED problem and provided supporting theoretical guarantees. Zhang et al. [45] propose an early interaction model, using hypergraphs to learn higher order node similarity. Each hypergraph convolution contains a subgraph matching module to learn cross graph similarity. Qin et al. [31] trained a slower attention-based network on multi-level features from a GNN and distilled its knowledge into a faster student model. Roy et al. [35] used the GumbelSinkhorn operator as a differentiable gadget to compute alignments in a backpropagation-friendly fashion and also demonstrated the utility of computing alignments for edges instead of nodes.

## C   Broader Impact

This work can be directly applied to numerous practical applications, such as drug discovery and circuit design, which are enormously beneficial for the society and continue to garner interest from researchers and practitioners worldwide. The ideas introduced in this paper have benefitted from and can benefit the information retrieval community as well, beyond the domain of graphs. However, malicious parties could use this technology for deceitful purposes, such as identifying and targeting specific social circles on online social networks (which can be represented as graphs). Such pros and cons are characteristic of every scientific study and the authors consider the positives to far outweigh the negatives.

## D   Network architecture of different components of IsoNet++

IsoNet++ models consist of three components - an encoder, a message-passing network and a node/edge aligner. We provide details about each of these components below. For convenience, we represent a linear layer with input dimension $a$ and output dimension $b$ as $\text{Linear}(a, b)$ and a linear-ReLU-linear network with $\text{Linear}(a, b)$, $\text{Linear}(b, c)$ layers with ReLU activation in the middle as $\text{LRL}(a, b, c)$.

### D.1   Encoder

The encoder transforms input node/edge features before they are fed into the message-passing network. For models centred around node alignment like IsoNet++ (Node), the encoder refers to $\text{Init}_\theta$ and is implemented as a $\text{Linear}(1, 10)$ layer. The edge vectors are not encoded and passed as-is down to the message-passing network. For edge-based models like IsoNet++ (Edge), the encoder refers to both $\text{Init}_{\theta,\text{node}}$ and $\text{Init}_{\theta,\text{edge}}$, which are implemented as $\text{Linear}(1, 10)$ and $\text{Linear}(1, 20)$ layers respectively.

### D.2   GNN

Within the message-passing framework, we use node embeddings of size $\dim_h = 10$ and edge embeddings of size $\dim_m = 20$. We specify each component of the GNN below.

- $\text{inter}_\theta$ combines the representation of the current node/edge ($h_\bullet$) with that from the other graph, which are together fed to the network by concatenation. For node-based and edge-based models, it is implemented as $\text{LRL}(20, 20, 10)$ and $\text{LRL}(40, 40, 20)$ networks respectively. In particular, we ensure that the input dimension is twice the size of the output dimension, which in turn equals the intermediate embedding dimension $\dim(z)$.
- $\text{msg}_\theta$ is used to compute messages by combining intermediate embeddings $z_\bullet$ of nodes across an edge with the representation of that edge. For node-based models, the edge vector is a fixed vector of size 1 while the intermediate node embeddings $z_\bullet$ are vectors of dimension 10, resulting in the network being a $\text{Linear}(21, 20)$ layer. For edge-based models, the edge embedding is the $m$ vector of size 20 which requires $\text{msg}_\theta$ to be a $\text{Linear}(40, 20)$ layer. Note that the message-passing network is applied twice, once to the ordered pair $(u, v)$ and then to $(v, u)$ and the outputs thus obtained are added up. This is to ensure node order invariance for undirected edges by design.
- $\text{comb}_\theta$ combines the representation of a node $z_\bullet$ with aggregated messages received by it from all its neighbors. It is modelled as a GRU where the node representation (of size 10) is the initial hidden state and the aggregated message vector (of size 20) is the only element of an input sequence which updates the hidden state to give us the final node embedding $h_\bullet$.

### D.3 Node aligner

The node aligner takes as input two sets of node vectors $\boldsymbol{H}^{(q)} \in \mathbb{R}^{n \times 10}$ and $\boldsymbol{H}^{(c)} \in \mathbb{R}^{n \times 10}$ representing $G_q$ and $G_c$ respectively. $n$ refers to the number of nodes in the corpus graph (the query graph is padded to meet this node count). We use $\mathrm{LRL}_\phi$ as a $\mathrm{LRL}(10, 16, 16)$ network (refer Eq. 11).

### D.4 Edge aligner

The design of the edge aligner is similar to the node aligner described above in Section D.3, except that its inputs are sets of edge vectors $\boldsymbol{M}^{(q)} \in \mathbb{R}^{e \times 20}$ and $\boldsymbol{M}^{(c)} \in \mathbb{R}^{e \times 20}$. $e$ refers to the number of edges in the corpus graph (the query graph is padded to meet this edge count). We use $\mathrm{LRL}_\phi$ as a $\mathrm{LRL}(20, 16, 16)$ network (refer Eq. 21).

### D.5 GumbelSinkhorn **operator**

The GumbelSinkhorn operator consists of the following operations -

$$\boldsymbol{D}_0 = \exp(\boldsymbol{D}_{\mathrm{in}}/\tau) \tag{26}$$
$$\boldsymbol{D}_{t+1} = \mathrm{RowNorm}\,(\mathrm{ColumnNorm}(\boldsymbol{D}_t)) \tag{27}$$
$$\boldsymbol{D}_{\mathrm{out}} = \lim_{t \to \infty} \boldsymbol{D}_t \tag{28}$$

The matrix $\boldsymbol{D}_{\mathrm{out}}$ obtained after this set of operations will be a doubly-stochastic matrix. The input $\boldsymbol{D}_{\mathrm{in}}$ in our case is the matrix containing the dot product of the node/edge embeddings of the query and corpus graphs respectively. $\tau$ represents the temperature and is fixed to $0.1$ in all our experiments.

**Theorem** Equation 11 results in a permutation matrix that is row-equivariant (column-) to the shuffling of nodes in $G_q$ ($G_c$).

**Proof** To prove the equivariance of Eq. 11, we need to show that given a shuffling (permutation) of query nodes $Z \in \Pi_n$ which modifies the node embedding matrix to $Z\dot{\boldsymbol{H}}_{t,K}^{(q)}$, the resulting output of said equation would change to $Z\boldsymbol{P}_t$. Below, we consider any matrices with $Z$ in the suffix as being an intermediate expression in the computation of $\mathrm{NodeAlignerRefinement}_\phi(Z\boldsymbol{H}_{t,K}^{(q)}, \boldsymbol{H}_{t,K}^{(c)})$.

It is easy to observe that the operators $\mathrm{LRL}_\phi$ (a linear-ReLU-linear network applied to a matrix), RowNorm, ColumnNorm and element-wise exponentiation ($\exp$), division are all permutation-equivariant since a shuffling of the vectors fed into these will trivially result in the output vectors getting shuffled in the same order. Thus, we get the following sequence of operations

$$\boldsymbol{D}_{\mathrm{in},Z} = \mathrm{LRL}_\phi(Z\boldsymbol{H}_{t,K}^{(q)})\,\mathrm{LRL}_\phi(\boldsymbol{H}_{t,K}^{(c)})^\top = Z \cdot \mathrm{LRL}_\phi(\boldsymbol{H}_{t,K}^{(q)})\,\mathrm{LRL}_\phi(\boldsymbol{H}_{t,K}^{(c)})^\top \boldsymbol{D}_{\mathrm{in}} = Z\boldsymbol{D}_{\mathrm{in}} \tag{29}$$

$\boldsymbol{D}_{0,Z}$ equals $\exp(\boldsymbol{D}_{\mathrm{in},Z}/\tau)$, which according to above equation would lead to $\boldsymbol{D}_{0,Z} = Z\boldsymbol{D}_0$. We can then inductively show using Eq. 27 and the equivariance of row/column normalization, assuming the following holds till $t$, that

$$\boldsymbol{D}_{t+1,Z} = \mathrm{RowNorm}\,(\mathrm{ColumnNorm}(\boldsymbol{D}_{t,Z})) = \mathrm{RowNorm}\,(\mathrm{ColumnNorm}(Z\boldsymbol{D}_t)) \tag{30}$$
$$= \mathrm{RowNorm}\,(Z \cdot \mathrm{ColumnNorm}(\boldsymbol{D}_t)) = Z \cdot \mathrm{RowNorm}\,(\mathrm{ColumnNorm}(\boldsymbol{D}_t)) = Z\boldsymbol{D}_{t+1} \tag{31}$$

The above equivariance would also hold in the limit, resulting in the doubly stochastic matrix $\boldsymbol{D}_{\mathrm{out},Z} = Z\boldsymbol{D}_{\mathrm{out}}$, which concludes the proof. ∎
A similar proof can be followed to show column equivariance for a shuffling in the corpus nodes.

## E Variants of our models and GMN, used in the experiments

### E.1 Multi-round refinement of IsoNet++ (Node) for the corpus graph

- Initialize:

$$\boldsymbol{h}_0^{(c)}(u') = \mathrm{Init}_\theta(\mathrm{feature}(u')), \tag{32}$$

- Update the GNN embeddings as follows:

$$\boldsymbol{z}_{t+1,k}^{(c)}(u') = \mathrm{inter}_\theta\left(\boldsymbol{h}_{t+1,k}^{(c)}(u'), \sum_{u \in V_q} \boldsymbol{h}_{t,k}^{(q)}(u)\boldsymbol{P}_t^\top[u', u]\right), \tag{33}$$

$$\boldsymbol{h}_{t+1,k+1}^{(c)}(u') = \mathrm{comb}_\theta\left(\boldsymbol{z}_{t+1,k}^{(c)}(u'), \sum_{v' \in \mathrm{nbr}(u')} \mathrm{msg}_\theta(\boldsymbol{z}_{t+1,k}^{(c)}(u'), \boldsymbol{z}_{t+1,k}^{(c)}(v'))\right) \tag{34}$$

## E.2   Eager update for IsoNet++ (Edge)

- Initialize:

$$h_0^{(q)}(u) = \text{Init}_{\theta,\text{node}}(\text{feature}(u)), \tag{35}$$

$$m_0^{(q)}(e) = \text{Init}_{\theta,\text{edge}}(\text{feature}(e)), \tag{36}$$

- The edge alignment is updated across layers. $S_0$ is set to a matrix of zeros. For $k > 0$, the following equation is used:

$$S_k = \text{EdgeAlignerRefinement}_\phi\left(M_k^{(q)}, M_k^{(c)}\right) \tag{37}$$

$$= \text{GumbelSinkhorn}\left(\text{LRL}_\phi(M_k^{(q)})\,\text{LRL}_\phi(M_k^{(c)})^\top\right) \tag{38}$$

- We update the GNN node and edge embeddings as follows:

$$z_k^{(q)}(e) = \text{inter}_\theta\left(m_k^{(q)}(e), \textstyle\sum_{e' \in E_c} m_k^{(c)}(e') S_k[e, e']\right), \tag{39}$$

$$h_{k+1}^{(q)}(u) = \text{comb}_\theta\left(h_k^{(q)}(u), \textstyle\sum_{a \in \text{nbr}(u)} \text{msg}_\theta(h_k^{(q)}(u), h_k^{(q)}(a), z_k^{(q)}((u,a)))\right) \tag{40}$$

$$m_{k+1}^{(q)}((u,v)) = \text{msg}_\theta(h_{k+1}^{(q)}(u), h_{k+1}^{(q)}(v), z_k^{(q)}((u,v))) \tag{41}$$

## E.3   Node partner (with additional MLP) variant of IsoNet++ (Node)

Here, we update node embeddings as follows:

$$h_{t+1,k+1}^{(q)}(u) = \text{comb}_\theta\left(z_{t+1,k}^{(q)}(u), \textstyle\sum_{v \in \text{nbr}(u)} \underbrace{\text{msg}_\theta(h_{t+1,k}^{(q)}(u), h_{t+1,k}^{(q)}(v))}_{z \text{ is replaced with } h}\right) \tag{42}$$

Here, $z_{t+1,k}^{(q)}(u)$ is computed as Eq. (12), where $\text{inter}_\theta$ is an MLP network. In contrast to Eq. (13), here, $z_{t+1,k}^{(q)}(u), z_{t+1,k}^{(q)}(v)$ are not fed into the message passing layer. Hence, in the message passing layer, we do not capture the signals from the partners of $u$ and $v$ in $G_c$. Only signals from partners of $u$ are captured through $z_{t+1,k}^{(q)}(u)$ in the first argument.

## E.4   Node pair partner (msg only) variant of IsoNet++ (Node)

We change the GNN update equation as follows:

$$h_{t+1,k+1}^{(q)}(u) = \text{comb}_\theta\left(\underbrace{h_{t+1,k}^{(q)}(u)}_{z \text{ is replaced with } h}, \textstyle\sum_{v \in \text{nbr}(u)} \text{msg}_\theta(z_{t+1,k}^{(q)}(u), z_{t+1,k}^{(q)}(v))\right) \tag{43}$$

Node pair partner interaction takes place because, we feed $z$ from Eq. (12) into the message passing layer. However, we use $h$ in the first argument, instead of $z$.

# F Additional details about experimental setup

## F.1 Datasets

We use six datasets from the TUDatasets collection [27] for benchmarking our methods with respect to existing baselines. Lou et al. [23] devised a method to sample query and corpus graphs from the graphs present in these datasets to create their training data. We adopt it for the task of subgraph matching. In particular, we choose a node $u \in G$ as the center of a Breadth First Search (BFS) and run the algorithm till $|V|$ nodes are traversed, where the range of $|V|$ is listed in Table 8 (refer to the Min and Max columns for $|V_q|$ and $|V_c|$). This process is independently performed for the query and corpus splits (with different ranges for graph size) to obtain 300 query graphs and 800 corpus graphs. The set of query graphs is split into train, validation and test splits of 180 (60%), 45 (15%) and 75 (25%) graphs respectively. Ground truth labels are computed for each query-corpus graph pair using the VF2 algorithm [9, 15, 23] implemented in the Networkx library. Various statistics about the datasets are listed in Table 8. pairs($y$) denotes the number of pairs in the dataset with gold label $y$, where $y \in \{0, 1\}$.

| | Mean $|V_q|$ | Min $|V_q|$ | Max $|V_q|$ | Mean $|E_q|$ | Mean $|V_c|$ | Min $|V_c|$ | Max $|V_c|$ | Mean $|E_c|$ | pairs(1) | pairs(0) | $\frac{\text{pairs}(1)}{\text{pairs}(0)}$ |
|---|---|---|---|---|---|---|---|---|---|---|---|
| AIDS | 11.61 | 7 | 15 | 11.25 | 18.50 | 17 | 20 | 18.87 | 41001 | 198999 | 0.2118 |
| Mutag | 12.91 | 6 | 15 | 13.27 | 18.41 | 17 | 20 | 19.89 | 42495 | 197505 | 0.2209 |
| FM | 11.73 | 6 | 15 | 11.35 | 18.30 | 17 | 20 | 18.81 | 40516 | 199484 | 0.2085 |
| FR | 11.81 | 6 | 15 | 11.39 | 18.32 | 17 | 20 | 18.79 | 39829 | 200171 | 0.2043 |
| MM | 11.80 | 6 | 15 | 11.37 | 18.36 | 17 | 20 | 18.79 | 40069 | 199931 | 0.2056 |
| MR | 11.87 | 6 | 15 | 11.49 | 18.32 | 17 | 20 | 18.78 | 40982 | 199018 | 0.2119 |

Table 8: Statistics for the 6 datasets borrowed from the TUDatasets collection [27]

## F.2 Baselines

**GraphSim, GOTSim, SimGNN, Neuromatch, GEN, GMN, IsoNet (Node), IsoNet (Edge)**: We utilized the code from official implementation of [35] [1]. Some *for loops* were vectorized to improve the running time of GMN.
**EGSC**: The official implementation [2] is refactored and integrated into our code.
**H2MN**: We use the official code from [3].
**GREED**: We use the official code from [4]. The model is adapted from the graph edit distance (GED) task to the subgraph isomorphism task, using a hinge scoring layer.

The number of parameters involved in all models (our methods and baselines) are reported in Table 9.

| | Number of parameters |
|---|---|
| GraphSim [2] | 3909 |
| GOTSim [11] | 304 |
| SimGNN [1] | 1671 |
| EGSC [31] | 3948 |
| H2MN [45] | 2974 |
| Neuromatch [23] | 3463 |
| GREED [32] | 1840 |
| GEN [22] | 1750 |
| GMN [22] | 2050 |
| IsoNet (Node) [35] | 1868 |
| IsoNet (Edge) [35] | 2028 |
| IsoNet++ (Node) | 2498 |
| IsoNet++ (Edge) | 4908 |

Table 9: Number of parameters for all models used in comparison

---

[1] https://github.com/Indradyumna/ISONET/
[2] https://github.com/canqin001/Efficient_Graph_Similarity_Computation
[3] https://github.com/cszhangzhen/H2MN
[4] https://github.com/idea-iitd/greed

## F.3  Calculation of Metrics: Mean Average Precision (MAP), HITS@K, Precision@K and Mean Reciprocal Rank (MRR)

Given a ranked list of corpus graphs $C = \{G_c\}$ for a test query $G_q$, sorted in the decreasing order of $\Delta_{\theta,\phi}(G_c|G_q)$, let us assume that the $c_+^{\text{th}}$ relevant graph is placed at position $\text{pos}(c_+) \in \{1, ..., |C|\}$ in the ranked list. Then Average Precision (AP) is computed as:

$$\text{AP}(q) = \frac{1}{|C_{q+}|} \sum_{c_+ \in [|C_{q+}|]} \frac{c_+}{\text{pos}(c_+)} \tag{44}$$

Mean average precision is defined as $\sum_{q \in Q} \text{AP}(q)/|Q|$.

Precision@$K(q) = \frac{1}{K}$ # relevant graphs corresponding to $G_q$ till rank $K$. Finally we report the mean of Precision@$K(q)$ across queries.

Reciprocal rank or $\text{RR}(q)$ is the inverse of the rank of the topmost relevant corpus graph corresponding to $G_q$ in the ranked list. Mean reciprocal rank (MRR) is average of $\text{RR}(q)$ across queries.

HITS@$K$ for a query $G_q$ is defined as the fraction of positively labeled corpus graphs that appear before the $K^{\text{th}}$ negatively labeled corpus graph. Finally, we report the average of HITS@$K$ across queries.

Note that HITS@K is a significantly aggressive metric compared to Precision@K and MRR, as can be seen in Tables 12 and 13.

## F.4  Details about hyperparameters

All models were trained using early stopping with MAP score on the validation split as a stopping criterion. For early stopping, we used a patience of $50$ with a tolerance of $10^{-4}$. We used the Adam optimizer with the learning rate as $10^{-3}$ and the weight decay parameter as $5 \cdot 10^{-4}$. We set batch size to $128$ and maximum number of epochs to $1000$.

**Seed Selection and Reproducibility**  Five integer seeds were chosen uniformly at random from the range $[0, 10^4]$ resulting in the set $\{1704, 4929, 7366, 7474, 7762\}$. IsoNet++ (Node), GMN and IsoNet (Edge) were trained on each of these 5 seeds for all 6 datasets. Note that these seeds do not control the training-dev-test splits but only control the initialization. Since the overall problem is non-convex, in principle, one should choose the best initial conditions leading to local minima. Hence, for all models, we choose the best seed, based on validation MAP score, is shown in Table 10.

| | AIDS | Mutag | FM | FR | MM | MR |
|---|---|---|---|---|---|---|
| GraphSim [2] | 7762 | 4929 | 7762 | 7366 | 4929 | 7474 |
| GOTSim [11] | 7762 | 7366 | 1704 | 7762 | 1704 | 7366 |
| SimGNN [1] | 7762 | 7474 | 1704 | 4929 | 4929 | 7762 |
| EGSC [31] | 4929 | 1704 | 7762 | 4929 | 4929 | 7366 |
| H2MN [45] | 7762 | 4929 | 7366 | 1704 | 4929 | 7474 |
| Neuromatch [23] | 7366 | 4929 | 7762 | 7762 | 1704 | 7366 |
| GREED [32] | 7762 | 1704 | 1704 | 7474 | 1704 | 1704 |
| GEN [22] | 1704 | 4929 | 7474 | 7762 | 1704 | 1704 |
| GMN [22] | 7366 | 4929 | 7366 | 7474 | 7474 | 7366 |
| IsoNet (Node) [35] | 7474 | 7474 | 7474 | 1704 | 4929 | 1704 |
| IsoNet (Edge) [35] | 7474 | 7474 | 7474 | 1704 | 4929 | 1704 |
| GMN [22] | 7366 | 4929 | 7366 | 7474 | 7474 | 7366 |
| IsoNet++ (Node) | 7762 | 7762 | 7474 | 7762 | 7762 | 7366 |

Table 10: Best seeds for all models. For IsoNet (Edge), GMN and IsoNet++ (Node), these are computed based on MAP score on the validation split at convergence. For other models, the identification occurs after 10 epochs of training.

IsoNet++ (Edge) and all ablations on top of IsoNet++ (Node) were trained using the best seeds for IsoNet++ (Node) (as in Tables 4, 5 and 16). Ablations of GMN were trained with the best GMN seeds.

For baselines excluding IsoNet (Edge), models were trained on all 5 seeds for 10 epochs and the MAP scores on the validation split were considered. Full training with early stopping was resumed only for

the best seed per dataset. This approach was adopted to reduce the computational requirements for benchmarking.

**Margin Selection**    For GraphSim, GOTSim, SimGNN, Neuromatch, GEN, GMN and IsoNet (Edge), we use the margins determined by Roy et al. [35] for each dataset. For IsoNet (Node), the margins prescribed for IsoNet (Edge) were used for standardization. For IsoNet++ (Node), IsoNet++ (Edge) and ablations, a fixed margin of $0.5$ is used.

Procedure for baselines **EGSC, GREED, H2MN**: They are trained on five seeds with a margin of 0.5 for 10 epochs and the best seed is chosen using the validation MAP score at this point. This seed is also used to train a model with a margin of 0.1 for 10 epochs. The better of these models, again using MAP score on the validation split, is identified and retrained till completion using early stopping.

|                   | AIDS | Mutag | FM  | FR  | MM  | MR  |
|-------------------|------|-------|-----|-----|-----|-----|
| GraphSim [2]      | 0.5  | 0.5   | 0.5 | 0.5 | 0.5 | 0.5 |
| GOTSim [11]       | 0.1  | 0.1   | 0.1 | 0.1 | 0.1 | 0.1 |
| SimGNN [1]        | 0.5  | 0.1   | 0.5 | 0.1 | 0.5 | 0.5 |
| EGSC [31]         | 0.1  | 0.5   | 0.1 | 0.5 | 0.1 | 0.5 |
| H2MN [45]         | 0.5  | 0.5   | 0.5 | 0.5 | 0.5 | 0.1 |
| Neuromatch [23]   | 0.5  | 0.5   | 0.5 | 0.5 | 0.5 | 0.5 |
| GREED [32]        | 0.5  | 0.5   | 0.5 | 0.5 | 0.5 | 0.5 |
| GEN [22]          | 0.5  | 0.5   | 0.5 | 0.5 | 0.5 | 0.5 |
| GMN [22]          | 0.5  | 0.5   | 0.5 | 0.5 | 0.5 | 0.5 |
| IsoNet (Node) [35]| 0.5  | 0.5   | 0.5 | 0.5 | 0.5 | 0.5 |
| IsoNet (Edge) [35]| 0.5  | 0.5   | 0.5 | 0.5 | 0.5 | 0.5 |

Table 11: Best margin for baselines used in comparison.

## F.5   Software and Hardware

All experiments were run with Python 3.10.13 and PyTorch 2.1.2. IsoNet++ (Node), IsoNet++ (Edge), GMN, IsoNet (Edge) and ablations on top of these were trained on Nvidia RTX A6000 (48 GB) GPUs while other baselines like GraphSim, GOTSim etc. were trained on Nvidia A100 (80 GB) GPUs.

As an estimate of training time, we typically spawn 3 training runs of IsoNet++ (Node) or IsoNet++ (Edge) on one Nvidia RTX A6000 GPU, each of which takes 300 epochs to conclude on average, with an average of 6-12 minutes per epoch. This amounts to 2 days of training. Overloading the GPUs by spawning 6 training runs per GPU increases the training time marginally to 2.5 days.

Additionally, we use wandb [5] to manage and monitor the experiments.

## F.6   License

GEN, GMN, GOTSim, GREED and EGSC are available under the MIT license, while SimGNN is public under the GNU license. The licenses for GraphSim, H2MN, IsoNet (Node), IsoNet (Edge), Neuromatch could not be identified. The authors were unable to identify the license of the TUDatasets repository [27], which was used to compile the 6 datasets used in this paper.

# G  Additional experiments

## G.1  Comparison against baselines

In Tables 12 and 13, we report the Mean Average Precision (MAP), HITS@20, MRR and Precision@20 scores for several baselines as well as the four approaches discussed in our paper - multi-layer and multi-round variants of IsoNet++ (Node) and IsoNet++ (Edge). Multi-round IsoNet++ (Edge) outperforms all other models with respect to all metrics, closely followed by multi-round IsoNet++ (Node) and multi-layer IsoNet++ (Edge) respectively. Among the baselines, IsoNet (Edge) is the best-performing model, closely followed by IsoNet (Node) and GMN.

For MRR, Precision@20, the comparisons are less indicative of the significant boost in performance obtained by IsoNet++, since these are not aggressive metrics from the point of view of information retrieval.

| Mean Average Precision (MAP) | | | | | | |
|---|---|---|---|---|---|---|
| | AIDS | Mutag | FM | FR | MM | MR |
| GraphSim [2] | $0.356 \pm 0.016$ | $0.472 \pm 0.027$ | $0.477 \pm 0.016$ | $0.423 \pm 0.019$ | $0.415 \pm 0.017$ | $0.453 \pm 0.018$ |
| GOTSim [11] | $0.324 \pm 0.015$ | $0.272 \pm 0.012$ | $0.355 \pm 0.014$ | $0.373 \pm 0.018$ | $0.323 \pm 0.015$ | $0.317 \pm 0.013$ |
| SimGNN [1] | $0.341 \pm 0.019$ | $0.283 \pm 0.012$ | $0.473 \pm 0.016$ | $0.341 \pm 0.015$ | $0.298 \pm 0.012$ | $0.379 \pm 0.015$ |
| EGSC [31] | $0.505 \pm 0.02$ | $0.476 \pm 0.022$ | $0.609 \pm 0.018$ | $0.607 \pm 0.019$ | $0.586 \pm 0.019$ | $0.58 \pm 0.018$ |
| H2MN [45] | $0.267 \pm 0.014$ | $0.276 \pm 0.012$ | $0.436 \pm 0.015$ | $0.412 \pm 0.016$ | $0.312 \pm 0.014$ | $0.243 \pm 0.008$ |
| Neuromatch [23] | $0.489 \pm 0.024$ | $0.576 \pm 0.029$ | $0.615 \pm 0.019$ | $0.559 \pm 0.024$ | $0.519 \pm 0.02$ | $0.606 \pm 0.021$ |
| GREED [32] | $0.472 \pm 0.021$ | $0.567 \pm 0.027$ | $0.558 \pm 0.02$ | $0.512 \pm 0.021$ | $0.546 \pm 0.021$ | $0.528 \pm 0.019$ |
| GEN [22] | $0.557 \pm 0.021$ | $0.605 \pm 0.028$ | $0.661 \pm 0.021$ | $0.575 \pm 0.02$ | $0.539 \pm 0.02$ | $0.631 \pm 0.018$ |
| GMN [22] | $0.622 \pm 0.02$ | $0.710 \pm 0.025$ | $0.730 \pm 0.018$ | $0.662 \pm 0.02$ | $0.655 \pm 0.02$ | $0.708 \pm 0.017$ |
| IsoNet (Node) [35] | $0.659 \pm 0.022$ | $0.697 \pm 0.026$ | $0.729 \pm 0.018$ | $0.68 \pm 0.022$ | $0.708 \pm 0.016$ | $0.738 \pm 0.017$ |
| IsoNet (Edge) [35] | $0.690 \pm 0.02$ | $0.706 \pm 0.026$ | $0.783 \pm 0.017$ | $0.722 \pm 0.02$ | $0.753 \pm 0.015$ | $0.774 \pm 0.016$ |
| multi-layer IsoNet++ (Node) | $0.756 \pm 0.019$ | $0.81 \pm 0.021$ | $0.859 \pm 0.015$ | $0.802 \pm 0.018$ | $0.827 \pm 0.015$ | $0.841 \pm 0.013$ |
| multi-layer IsoNet++ (Edge) | $0.795 \pm 0.018$ | $0.805 \pm 0.022$ | $0.883 \pm 0.013$ | $0.812 \pm 0.016$ | $0.862 \pm 0.013$ | $0.886 \pm 0.011$ |
| multi-round IsoNet++ (Node) | $0.825 \pm 0.016$ | $0.851 \pm 0.018$ | $0.888 \pm 0.012$ | $0.855 \pm 0.015$ | $0.838 \pm 0.015$ | $0.874 \pm 0.011$ |
| multi-round IsoNet++ (Edge) | $0.847 \pm 0.016$ | $0.858 \pm 0.019$ | $0.902 \pm 0.012$ | $0.875 \pm 0.014$ | $0.902 \pm 0.01$ | $0.902 \pm 0.01$ |

| HITS@20 | | | | | | |
|---|---|---|---|---|---|---|
| | AIDS | Mutag | FM | FR | MM | MR |
| GraphSim [2] | $0.145 \pm 0.011$ | $0.257 \pm 0.027$ | $0.261 \pm 0.015$ | $0.227 \pm 0.015$ | $0.212 \pm 0.014$ | $0.23 \pm 0.015$ |
| GOTSim [11] | $0.112 \pm 0.011$ | $0.088 \pm 0.009$ | $0.147 \pm 0.011$ | $0.166 \pm 0.014$ | $0.119 \pm 0.011$ | $0.116 \pm 0.011$ |
| SimGNN [1] | $0.138 \pm 0.016$ | $0.087 \pm 0.008$ | $0.235 \pm 0.015$ | $0.155 \pm 0.013$ | $0.111 \pm 0.009$ | $0.160 \pm 0.013$ |
| EGSC [31] | $0.267 \pm 0.023$ | $0.243 \pm 0.02$ | $0.364 \pm 0.02$ | $0.382 \pm 0.024$ | $0.348 \pm 0.023$ | $0.325 \pm 0.021$ |
| H2MN [45] | $0.076 \pm 0.009$ | $0.084 \pm 0.007$ | $0.200 \pm 0.012$ | $0.189 \pm 0.013$ | $0.119 \pm 0.011$ | $0.069 \pm 0.004$ |
| Neuromatch [23] | $0.262 \pm 0.025$ | $0.376 \pm 0.034$ | $0.389 \pm 0.022$ | $0.350 \pm 0.025$ | $0.282 \pm 0.019$ | $0.385 \pm 0.025$ |
| GREED [32] | $0.245 \pm 0.025$ | $0.371 \pm 0.034$ | $0.316 \pm 0.027$ | $0.287 \pm 0.019$ | $0.311 \pm 0.024$ | $0.277 \pm 0.023$ |
| GEN [22] | $0.321 \pm 0.026$ | $0.429 \pm 0.035$ | $0.448 \pm 0.03$ | $0.368 \pm 0.026$ | $0.292 \pm 0.024$ | $0.391 \pm 0.025$ |
| GMN [22] | $0.397 \pm 0.029$ | $0.544 \pm 0.035$ | $0.537 \pm 0.027$ | $0.45 \pm 0.027$ | $0.423 \pm 0.025$ | $0.49 \pm 0.026$ |
| IsoNet (Node) [35] | $0.438 \pm 0.028$ | $0.509 \pm 0.034$ | $0.525 \pm 0.026$ | $0.475 \pm 0.03$ | $0.493 \pm 0.023$ | $0.532 \pm 0.025$ |
| IsoNet (Edge) [35] | $0.479 \pm 0.029$ | $0.529 \pm 0.035$ | $0.613 \pm 0.026$ | $0.538 \pm 0.029$ | $0.571 \pm 0.023$ | $0.601 \pm 0.027$ |
| multi-layer IsoNet++ (Node) | $0.57 \pm 0.029$ | $0.672 \pm 0.033$ | $0.744 \pm 0.027$ | $0.657 \pm 0.031$ | $0.68 \pm 0.025$ | $0.707 \pm 0.024$ |
| multi-layer IsoNet++ (Edge) | $0.626 \pm 0.029$ | $0.671 \pm 0.035$ | $0.775 \pm 0.026$ | $0.67 \pm 0.028$ | $0.743 \pm 0.024$ | $0.776 \pm 0.021$ |
| multi-round IsoNet++ (Node) | $0.672 \pm 0.027$ | $0.732 \pm 0.03$ | $0.797 \pm 0.024$ | $0.737 \pm 0.026$ | $0.702 \pm 0.025$ | $0.755 \pm 0.022$ |
| multi-round IsoNet++ (Edge) | $0.705 \pm 0.028$ | $0.749 \pm 0.032$ | $0.813 \pm 0.023$ | $0.769 \pm 0.026$ | $0.809 \pm 0.019$ | $0.803 \pm 0.02$ |

Table 12: Replication of Table 3 with standard error. Comparison of the two variants of IsoNet++ (IsoNet++ (Node) and IsoNet++ (Edge)) against all the state-of-the-art graph retrieval methods, across all six datasets. Performance is measured in terms average precision MAP and HITS@20. In all cases, we used 60% training, 15% validation and 25% test sets. The first five methods apply a neural network on the fused graph-pair representations. The next six methods apply asymmetric hinge distance between the query and corpus embeddings similar to our method. The numbers with green and yellow indicate the best, second best method respectively, whereas the numbers with blue indicate the best method among the baselines. (MAP values for IsoNet++ (Edge) across FM, MM and MR are verified to be not exactly same, but they take the same value until the third decimal).

| Mean Reciprocal Rank (MRR) | | | | | |
| --- | --- | --- | --- | --- | --- |
| | AIDS | Mutag | FM | FR | MM | MR |
| GraphSim [2] | 0.71 ±0.039 | 0.795 ±0.037 | 0.885 ±0.029 | 0.817 ±0.032 | 0.818 ±0.034 | 0.789 ±0.037 |
| GOTSim [11] | 0.568 ±0.038 | 0.584 ±0.037 | 0.775 ±0.037 | 0.716 ±0.042 | 0.459 ±0.045 | 0.525 ±0.047 |
| SimGNN [1] | 0.533 ±0.038 | 0.644 ±0.043 | 0.866 ±0.031 | 0.753 ±0.038 | 0.669 ±0.04 | 0.638 ±0.046 |
| EGSC [31] | 0.894 ±0.026 | 0.75 ±0.041 | 0.943 ±0.021 | 0.909 ±0.023 | 0.904 ±0.025 | 0.932 ±0.022 |
| H2MN [45] | 0.46 ±0.047 | 0.565 ±0.042 | 0.822 ±0.035 | 0.817 ±0.034 | 0.386 ±0.039 | 0.62 ±0.041 |
| Neuromatch [23] | 0.823 ±0.035 | 0.855 ±0.035 | 0.88 ±0.028 | 0.929 ±0.022 | 0.87 ±0.027 | 0.895 ±0.026 |
| GREED [32] | 0.789 ±0.035 | 0.805 ±0.034 | 0.834 ±0.033 | 0.834 ±0.032 | 0.894 ±0.028 | 0.759 ±0.039 |
| GEN [22] | 0.865 ±0.028 | 0.895 ±0.029 | 0.889 ±0.026 | 0.878 ±0.028 | 0.814 ±0.034 | 0.878 ±0.026 |
| GMN [22] | 0.877 ±0.027 | 0.923 ±0.023 | 0.949 ±0.019 | 0.947 ±0.019 | 0.928 ±0.023 | 0.922 ±0.022 |
| IsoNet (Node) [35] | 0.916 ±0.024 | 0.887 ±0.029 | 0.977 ±0.013 | 0.954 ±0.018 | 0.956 ±0.018 | 0.954 ±0.018 |
| IsoNet (Edge) [35] | 0.949 ±0.02 | 0.926 ±0.026 | 0.973 ±0.013 | 0.956 ±0.018 | 0.98 ±0.011 | 0.948 ±0.019 |
| multi-layer IsoNet++ (Node) | 0.956 ±0.018 | 0.954 ±0.018 | 1.0 ±0.0 | 0.978 ±0.013 | 0.98 ±0.011 | 1.0 ±0.0 |
| multi-layer IsoNet++ (Edge) | 0.984 ±0.011 | 0.976 ±0.014 | 0.991 ±0.009 | 0.987 ±0.009 | 0.987 ±0.009 | 0.993 ±0.007 |
| multi-round IsoNet++ (Node) | 0.993 ±0.007 | 0.971 ±0.014 | 1.0 ±0.0 | 0.993 ±0.007 | 0.993 ±0.007 | 0.993 ±0.007 |
| multi-round IsoNet++ (Edge) | 1.0 ±0.0 | 0.983 ±0.012 | 0.991 ±0.009 | 1.0 ±0.0 | 1.0 ±0.0 | 1.0 ±0.0 |

| Precision@20 | | | | | |
| --- | --- | --- | --- | --- | --- |
| | AIDS | Mutag | FM | FR | MM | MR |
| GraphSim [2] | 0.474 ±0.025 | 0.577 ±0.033 | 0.679 ±0.023 | 0.617 ±0.028 | 0.604 ±0.028 | 0.638 ±0.026 |
| GOTSim [11] | 0.386 ±0.024 | 0.325 ±0.021 | 0.479 ±0.027 | 0.519 ±0.03 | 0.409 ±0.027 | 0.421 ±0.03 |
| SimGNN [1] | 0.44 ±0.026 | 0.33 ±0.022 | 0.626 ±0.026 | 0.471 ±0.029 | 0.414 ±0.026 | 0.512 ±0.032 |
| EGSC [31] | 0.646 ±0.023 | 0.608 ±0.034 | 0.79 ±0.022 | 0.766 ±0.021 | 0.739 ±0.023 | 0.74 ±0.021 |
| H2MN [45] | 0.28 ±0.026 | 0.34 ±0.023 | 0.587 ±0.024 | 0.563 ±0.026 | 0.399 ±0.028 | 0.308 ±0.017 |
| Neuromatch [23] | 0.615 ±0.03 | 0.689 ±0.032 | 0.809 ±0.022 | 0.725 ±0.027 | 0.694 ±0.027 | 0.751 ±0.023 |
| GREED [32] | 0.591 ±0.024 | 0.661 ±0.03 | 0.689 ±0.026 | 0.642 ±0.028 | 0.699 ±0.028 | 0.624 ±0.029 |
| GEN [22] | 0.674 ±0.024 | 0.721 ±0.03 | 0.783 ±0.023 | 0.678 ±0.022 | 0.64 ±0.027 | 0.759 ±0.021 |
| GMN [22] | 0.751 ±0.022 | 0.82 ±0.023 | 0.852 ±0.02 | 0.809 ±0.019 | 0.783 ±0.022 | 0.832 ±0.018 |
| IsoNet (Node) [35] | 0.791 ±0.022 | 0.803 ±0.029 | 0.866 ±0.018 | 0.803 ±0.022 | 0.844 ±0.015 | 0.863 ±0.016 |
| IsoNet (Edge) [35] | 0.822 ±0.022 | 0.812 ±0.028 | 0.896 ±0.016 | 0.851 ±0.017 | 0.877 ±0.014 | 0.875 ±0.017 |
| multi-layer IsoNet++ (Node) | 0.873 ±0.018 | 0.897 ±0.018 | 0.935 ±0.012 | 0.917 ±0.012 | 0.93 ±0.013 | 0.931 ±0.012 |
| multi-layer IsoNet++ (Edge) | 0.905 ±0.015 | 0.883 ±0.021 | 0.958 ±0.01 | 0.93 ±0.01 | 0.953 ±0.01 | 0.976 ±0.005 |
| multi-round IsoNet++ (Node) | 0.932 ±0.012 | 0.943 ±0.011 | 0.957 ±0.01 | 0.961 ±0.008 | 0.949 ±0.011 | 0.963 ±0.008 |
| multi-round IsoNet++ (Edge) | 0.946 ±0.012 | 0.931 ±0.014 | 0.973 ±0.007 | 0.963 ±0.008 | 0.98 ±0.005 | 0.987 ±0.003 |

Table 13: MRR and Precision@20 of corresponding models from Table 3 with standard error. Comparison of the two variants of IsoNet++ (IsoNet++ (Node) and IsoNet++ (Edge)) against all the state-of-the-art graph retrieval methods, across all six datasets. Performance is measured in terms MRR and Precision@20. In all cases, we used 60% training, 15% validation and 25% test sets. The first five methods apply a neural network on the fused graph-pair representations. The next six methods apply asymmetric hinge distance between the query and corpus embeddings similar to our method. The numbers with green and yellow indicate the best, second best method respectively, whereas the numbers with blue indicate the best method among the baselines.

## G.2 HITS@20, MRR and Precision@20 for multi-round IsoNet++ and multi-layer IsoNet++

Table 14 compares multi-round and multi-layer IsoNet++ with respect to different metrics. We observe that multi-round IsoNet++ outperforms multi-layer IsoNet++ by a significant margin when it comes to all metrics, both when the models are node-based or edge-based. This reinforces the observations from MAP scores noted earlier in Table 4. Note that a minor exception occurs for MRR but the scores are already so close to 1 that this particular metric can be discounted and our key observation above still stands.

## G.3 Refinement of alignment matrix across rounds and layers in multi-round IsoNet++ and multi-layer IsoNet++

The node (edge) alignment calculated after round $t$ is denoted as $P_t$ ($S_t$). We accumulate such alignments across multiple rounds. This also includes $P_T$ ($S_T$) which is used to compute the relevance distance in Eq. 14 (Eq. 25). We wish to compare the predicted alignments with ground

| | HITS@20 | | | | | |
|---|---|---|---|---|---|---|
| | AIDS | Mutag | FM | FR | MM | MR |
| **Node** Multi-layer | 0.57 | 0.672 | 0.744 | 0.657 | 0.68 | 0.707 |
| **Node** Multi-round | 0.672 | 0.732 | 0.797 | 0.737 | 0.702 | 0.755 |
| **Edge** Multi-layer | 0.626 | 0.671 | 0.775 | 0.67 | 0.743 | 0.776 |
| **Edge** Multi-round | 0.705 | 0.749 | 0.813 | 0.769 | 0.809 | 0.803 |

| | Mean Reciprocal Rank (MRR) | | | | | |
|---|---|---|---|---|---|---|
| | AIDS | Mutag | FM | FR | MM | MR |
| **Node** Multi-layer | 0.956 | 0.954 | 1.0 | 0.978 | 0.98 | 1.0 |
| **Node** Multi-round | 0.993 | 0.971 | 1.0 | 0.993 | 0.993 | 0.993 |
| **Edge** Multi-layer | 0.984 | 0.976 | 0.991 | 0.987 | 0.987 | 0.993 |
| **Edge** Multi-round | 1.0 | 0.983 | 0.991 | 1.0 | 1.0 | 1.0 |

| | Precision@20 | | | | | |
|---|---|---|---|---|---|---|
| | AIDS | Mutag | FM | FR | MM | MR |
| **Node** Multi-layer | 0.873 | 0.897 | 0.935 | 0.917 | 0.93 | 0.931 |
| **Node** Multi-round | 0.932 | 0.943 | 0.957 | 0.961 | 0.949 | 0.963 |
| **Edge** Multi-layer | 0.905 | 0.883 | 0.958 | 0.93 | 0.953 | 0.976 |
| **Edge** Multi-round | 0.946 | 0.931 | 0.973 | 0.963 | 0.98 | 0.987 |

Table 14: Multi-round vs. multi-layer refinement. First and the last two rows of each table report HITS@20, MRR and Precision@20 for IsoNet++ (Node) and IsoNet++ (Edge) respectively. Rows colored green and yellow indicate the best and second best methods respectively.

truth alignments. We expect our final alignment matrix $P_t$ ($S_t$) to be one of them. We determine the closest ground truth matrices $P^*$ and $S^*$ by computing $\max_P \text{Tr}(P_T^\top P)$ and $\max_S \text{Tr}(S_T^\top S)$ for IsoNet++ (Node) and IsoNet++ (Edge) respectively. We now use the closest ground-truth alignment $P^*$, to compute $\text{Tr}(P_t^\top P^*)$ for $t \in [T]$. For each $t$, we plot a histogram with bin width 0.1 that denotes the density estimate $\text{p}(\text{Tr}(P_t^\top P^*))$. The same procedure is adopted for edges, with $S^*$ used instead of $P^*$. The histograms are depicted in Figure 15. We observe that the plots shift rightward with increasing $t$. The frequency of graph pairs with misaligned $P_t$ ($S_t$) decreases with rounds $t$ while that with well-aligned $P_t$ ($S_t$) increases.

Here, we also study alignments obtained through multi-layer refinement. We adopt the same procedure as in Section G.3. One key difference is that the node/edge alignments are computed after every layer $k$ and are accumulated across layers $k \in [K]$. In Figure 15, we observe that the plots, in general, shift rightward with increasing $k$. The frequency of graph pairs with misaligned $P_t$ ($S_t$) decreases with rounds $k$ while that with well-aligned $P_k$ ($S_k$) increases.

### G.4 Comparison across alternatives of multi-layer IsoNet++ (Node) and multi-round IsoNet++ (Node)

In Table 16, we compare different alternatives to the multi-round and multi-layer variants of IsoNet++ (Node). In particular, we consider four alternatives - Node partner (equation shown in Section 4), Node partner (with additional MLP) [Appendix E.3], Node pair partner (msg only) [Appendix E.4] and IsoNet++ (Node). We observe that for all metrics, IsoNet++ (Node) and Node pair partner (msg only) dominate the other alternatives in most cases. This highlights the importance of node pair partner interaction for determining the subgraph isomorphism relationship between two graphs. For the multi-round variant, IsoNet++ (Node) outperforms Node pair partner (msg only) in four of the datasets and is comparable / slightly worse in the other two. Once again, comparisons based on MRR break down because it does not cause a strong differentiation between the approaches.

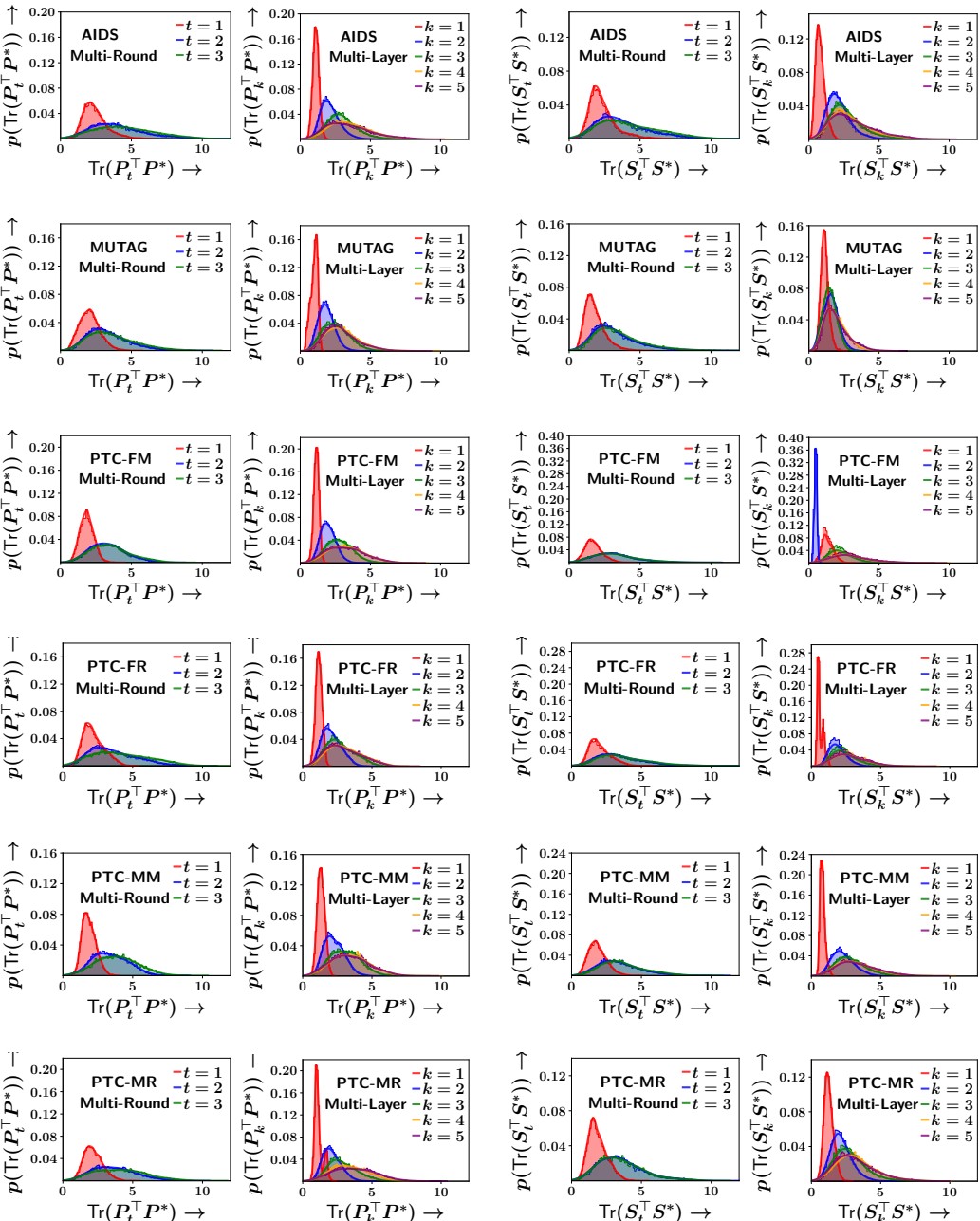

Figure 15: Similar to Figure 6, we plot empirical probability density of $p(\text{Tr}(\boldsymbol{P}_t^{\top}\boldsymbol{P}^*))$ and $p(\text{Tr}(\boldsymbol{S}_t^{\top}\boldsymbol{S}^*))$ for different values of $t$ lazy multi round updates and $p(\text{Tr}(\boldsymbol{P}_k^{\top}\boldsymbol{P}^*))$ and $p(\text{Tr}(\boldsymbol{S}_k^{\top}\boldsymbol{S}^*))$ for different values of $k$ for eager multi layer updates. The first (last) two plots in the left (right) of each row are for multi-round IsoNet++ (Node) (multi-round IsoNet++ (Edge)).

**Mean Average Precision (MAP)**

| | | AIDS | Mutag | FM | FR | MM | MR |
|---|---|---|---|---|---|---|---|
| Multi-Layer | Node partner (with additional MLP) | 0.692 | 0.782 | 0.822 | 0.776 | 0.777 | 0.803 |
| | Node pair partner (msg only) | 0.765 | 0.792 | 0.876 | 0.823 | 0.843 | 0.848 |
| | Node partner | 0.668 | 0.783 | 0.821 | 0.752 | 0.753 | 0.794 |
| | IsoNet++ (Node) | 0.756 | 0.81 | 0.859 | 0.802 | 0.827 | 0.841 |
| Multi-Round | Node partner (with additional MLP) | 0.815 | 0.844 | 0.868 | 0.852 | 0.818 | 0.858 |
| | Node pair partner (msg only) | 0.818 | 0.833 | 0.897 | 0.831 | 0.852 | 0.871 |
| | Node partner | 0.776 | 0.829 | 0.851 | 0.819 | 0.844 | 0.84 |
| | IsoNet++ (Node) | 0.825 | 0.851 | 0.888 | 0.855 | 0.838 | 0.874 |

**HITS@20**

| | | AIDS | Mutag | FM | FR | MM | MR |
|---|---|---|---|---|---|---|---|
| Multi-Layer | Node partner (with additional MLP) | 0.479 | 0.634 | 0.677 | 0.611 | 0.608 | 0.64 |
| | Node pair partner (msg only) | 0.577 | 0.651 | 0.775 | 0.682 | 0.719 | 0.703 |
| | Node partner | 0.433 | 0.639 | 0.678 | 0.58 | 0.571 | 0.624 |
| | IsoNet++ (Node) | 0.57 | 0.672 | 0.744 | 0.657 | 0.68 | 0.707 |
| Multi-Round | Node partner (with additional MLP) | 0.658 | 0.727 | 0.756 | 0.738 | 0.667 | 0.743 |
| | Node pair partner (msg only) | 0.671 | 0.717 | 0.807 | 0.696 | 0.728 | 0.753 |
| | Node partner | 0.603 | 0.702 | 0.736 | 0.686 | 0.721 | 0.695 |
| | IsoNet++ (Node) | 0.672 | 0.732 | 0.797 | 0.737 | 0.702 | 0.755 |

**Mean Reciprocal Rank (MRR)**

| | | AIDS | Mutag | FM | FR | MM | MR |
|---|---|---|---|---|---|---|---|
| Multi-Layer | Node partner (with additional MLP) | 0.909 | 0.941 | 0.965 | 0.964 | 0.966 | 0.984 |
| | Node pair partner (msg only) | 0.97 | 0.956 | 0.964 | 0.993 | 0.978 | 1.0 |
| | Node partner | 0.917 | 0.945 | 0.964 | 0.987 | 0.958 | 0.969 |
| | IsoNet++ (Node) | 0.956 | 0.954 | 1.0 | 0.978 | 0.98 | 1.0 |
| Multi-Round | Node partner (with additional MLP) | 0.987 | 0.944 | 0.993 | 0.987 | 0.963 | 0.983 |
| | Node pair partner (msg only) | 0.984 | 0.958 | 0.993 | 0.98 | 0.984 | 0.984 |
| | Node partner | 0.984 | 0.949 | 0.993 | 0.978 | 0.978 | 0.97 |
| | IsoNet++ (Node) | 0.993 | 0.971 | 1.0 | 0.993 | 0.993 | 0.993 |

**Precision@20**

| | | AIDS | Mutag | FM | FR | MM | MR |
|---|---|---|---|---|---|---|---|
| Multi-Layer | Node partner (with additional MLP) | 0.817 | 0.867 | 0.913 | 0.913 | 0.883 | 0.914 |
| | Node pair partner (msg only) | 0.871 | 0.886 | 0.957 | 0.937 | 0.927 | 0.937 |
| | Node partner | 0.799 | 0.866 | 0.919 | 0.877 | 0.873 | 0.885 |
| | IsoNet++ (Node) | 0.873 | 0.897 | 0.935 | 0.917 | 0.93 | 0.931 |
| Multi-Round | Node partner (with additional MLP) | 0.921 | 0.917 | 0.936 | 0.951 | 0.921 | 0.945 |
| | Node pair partner (msg only) | 0.923 | 0.913 | 0.969 | 0.951 | 0.957 | 0.957 |
| | Node partner | 0.875 | 0.921 | 0.933 | 0.942 | 0.939 | 0.941 |
| | IsoNet++ (Node) | 0.932 | 0.943 | 0.957 | 0.961 | 0.949 | 0.963 |

Table 16: Effect of node pair partner interaction in IsoNet++ (Node). Table shows the comparison of IsoNet++ (Node) with three different alternatives. The first table reports MAP values, second reports HITS@20, third reports MRR and fourth reports Precision@20. In each table, the first two rows report metrics for multi-layer refinement and the second two rows report metrics for multi-round refinement. Rows colored green and yellow indicate the best and second best methods in their respective sections.

## G.5   Comparison of GMN with IsoNet++ alternative for multi-layer and multi-round

In Table 17, we modify the GMN architecture to include node pair partner interaction in the message-passing layer. Based on the reported metrics, we observe that there is no substantial improvement upon including information from node pairs in GMN, which is driven by a non-injective mapping (attention). This indicates that injectivity of the doubly stochastic matrix in our formulation is crucial towards the boost in performance obtained from node pair partner interaction as well.

| | | AIDS | Mutag | FM | FR | MM | MR |
|---|---|---|---|---|---|---|---|
| | **Mean Average Precision (MAP)** | | | | | | |
| Multi-Layer | GMN | 0.622 | 0.71 | 0.73 | 0.662 | 0.655 | 0.708 |
| | Node pair partner | 0.579 | 0.732 | 0.74 | 0.677 | 0.641 | 0.713 |
| Multi-Round | GMN | 0.629 | 0.699 | 0.757 | 0.697 | 0.653 | 0.714 |
| | Node pair partner | 0.579 | 0.693 | 0.729 | 0.69 | 0.665 | 0.705 |

| | | AIDS | Mutag | FM | FR | MM | MR |
|---|---|---|---|---|---|---|---|
| | **HITS@20** | | | | | | |
| Multi-Layer | GMN | 0.397 | 0.544 | 0.537 | 0.45 | 0.423 | 0.49 |
| | Node pair partner | 0.346 | 0.567 | 0.551 | 0.476 | 0.411 | 0.5 |
| Multi-Round | GMN | 0.403 | 0.533 | 0.562 | 0.494 | 0.431 | 0.502 |
| | Node pair partner | 0.344 | 0.528 | 0.54 | 0.502 | 0.462 | 0.506 |

| | | AIDS | Mutag | FM | FR | MM | MR |
|---|---|---|---|---|---|---|---|
| | **Mean Reciprocal Rank (MRR)** | | | | | | |
| Multi-Layer | GMN | 0.877 | 0.923 | 0.949 | 0.947 | 0.928 | 0.922 |
| | Node pair partner | 0.827 | 0.897 | 0.958 | 0.877 | 0.918 | 0.92 |
| Multi-Round | GMN | 0.905 | 0.862 | 0.958 | 0.956 | 0.906 | 0.921 |
| | Node pair partner | 0.811 | 0.901 | 0.907 | 0.908 | 0.964 | 0.92 |

| | | AIDS | Mutag | FM | FR | MM | MR |
|---|---|---|---|---|---|---|---|
| | **Precision@20** | | | | | | |
| Multi-Layer | GMN | 0.751 | 0.82 | 0.852 | 0.809 | 0.783 | 0.832 |
| | Node pair partner | 0.7 | 0.833 | 0.861 | 0.797 | 0.792 | 0.846 |
| Multi-Round | GMN | 0.753 | 0.795 | 0.885 | 0.829 | 0.792 | 0.842 |
| | Node pair partner | 0.694 | 0.794 | 0.847 | 0.835 | 0.802 | 0.825 |

Table 17: Effect of node pair partner interaction in GMN. The tables compare GMN with its IsoNet++ alternative. The first table reports MAP values, the second table reports HITS@20 values, the third table reports MRR values and the fourth table reports Precision@20. In each table, the first two rows report metrics for multi-layer refinement and the second two rows report metrics for multi-round refinement. Rows colored green and yellow indicate the best and second best methods according to the respective metrics.

### G.6 Variation of IsoNet++ (Node) and IsoNet++ (Edge) with different $T$ and $K$

In this section, we analyze the accuracy and inference time trade-off of multi-round lazy and multi-layer eager variants of IsoNet++ (Node) and IsoNet++ (Edge). In the following tables, we show the MAP and inference time. Additionally, we also analyze the trade-off of GMN and IsoNet (Edge). The $T, K$ parameters for different models are so chosen that they can be compared against each other while fixing the inference time to be roughly similar. For instance, multi-round lazy IsoNet++ (Node) with $T = 5, K = 5$ maps to multi-layer eager IsoNet++ (Node) with $K = 8$, allowing for a direct comparison of performance without caring much about different compute. Note that in below tables, models are listed in order of increasing inference time (i.e. increasing $K$ or $T$).

In tables 19 and 20, we show variations for multi-round lazy IsoNet++ (Node) for fixed $T$ and fixed $K$ respectively. We observe that with fixed $T$, increasing $K$ from 5 to 10 doesn't improve the model significantly. For fixed $K$, performance (in terms of MAP) improves notably when increasing $T$ from 3 to 5.

In table 21, we show variations for multi-layer eager IsoNet++ (Node) for varying $K$. We observe that except for a drop at $K = 7$, the performance of the model improves as we increase $K$. In fact, at $K = 8$, the performance is surprisingly good, even outperforming the similarly timed $T = 5, K = 5$ variant of lazy multi-round IsoNet++ (Node) on both AIDS and Mutag.

In tables 22 and 23, we compare variants of multi-round lazy IsoNet++ (Edge) with fixed $T$ and fixed $K$ respectively. We observe that when $T$ is fixed and $K$ is increased, the gain is marginal. We observe a significant gain When $K$ is fixed and $T$ is increased from 3 to 4.

In table 24, we study the trade-off for multi-layer eager IsoNet++ (Edge) for varying $K$. We observe that with increasing $K$, the performance continues to improve and peaks at $K = 8$. Note that even at this $K$, the performance of multi-layer eager IsoNet++ (Edge) is worse than a similarly timed variant ($T = 5, K = 5$) of multi-round IsoNet++ (Edge).

In table 25, we show variations for GMN for varying $K$. We observe marginal gains while increasing $K$. From $K = 10$ to $K = 12$, the performance drops.

In table 26, we show how performance varies for IsoNet (Edge) for varying $K$. We observe that the model does not improve with increasing $K$.

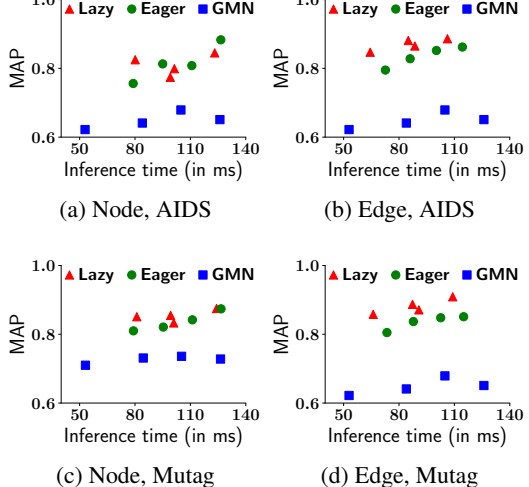

Figure 18: Trade off between MAP and inference time (batch size=128).

| Mean Average Precision (MAP) | | | | | | |
|---|---|---|---|---|---|---|
| | AIDS | Mutag | FM | FR | MM | MR |
| $T = 3, K = 5$ | 0.825 | 0.851 | 0.888 | 0.855 | 0.838 | 0.874 |
| $T = 3, K = 10$ | 0.774 | 0.855 | 0.898 | 0.811 | 0.855 | 0.882 |

| Inference time (in ms) | | | | | | |
|---|---|---|---|---|---|---|
| | AIDS | Mutag | FM | FR | MM | MR |
| $T = 3, K = 5$ | 80.11 | 80.99 | 81.01 | 81.24 | 80.94 | 80.25 |
| $T = 3, K = 10$ | 99.11 | 99.31 | 99.28 | 99.48 | 99.37 | 99.36 |

Table 19: MAP and inference time trade-off of variants of multi-round lazy IsoNet++ (Node) with fixed $T$. Rows colored green indicate the best $K$ according to the MAP score.

| Mean Average Precision (MAP) | | | | | | |
|---|---|---|---|---|---|---|
| | AIDS | Mutag | FM | FR | MM | MR |
| $T = 3, K = 5$ | 0.825 | 0.851 | 0.888 | 0.855 | 0.838 | 0.874 |
| $T = 4, K = 5$ | 0.799 | 0.833 | 0.892 | 0.858 | 0.867 | 0.891 |
| $T = 5, K = 5$ | 0.845 | 0.875 | 0.919 | 0.883 | 0.894 | 0.897 |

| Inference time (in ms) | | | | | | |
|---|---|---|---|---|---|---|
| | AIDS | Mutag | FM | FR | MM | MR |
| $T = 3, K = 5$ | 80.11 | 80.99 | 81.01 | 81.24 | 80.94 | 80.25 |
| $T = 4, K = 5$ | 101.33 | 100.99 | 100.95 | 100.46 | 100.59 | 100.87 |
| $T = 5, K = 5$ | 123.18 | 124.19 | 123.61 | 122.79 | 123.33 | 122.74 |

Table 20: MAP and inference time trade-off of variants of multi-round lazy IsoNet++ (Node) with fixed $K$. Rows colored green and yellow indicate the best and second best $T$ according to the MAP score.

| Mean Average Precision (MAP) | | |
|---|---|---|
| | AIDS | Mutag |
| $K = 5$ | 0.756 | 0.81 |
| $K = 6$ | 0.813 | 0.821 |
| $K = 7$ | 0.808 | 0.842 |
| $K = 8$ | 0.883 | 0.874 |

| Inference time (in ms) | | |
|---|---|---|
| | AIDS | Mutag |
| $K = 5$ | 79.02 | 79.15 |
| $K = 6$ | 94.99 | 95.33 |
| $K = 7$ | 110.78 | 111.09 |
| $K = 8$ | 126.48 | 126.6 |

Table 21: MAP and inference time trade-off of variants of multi-layer eager IsoNet++ (Node) with increasing $K$. Rows colored green and yellow indicate the best and second best $K$ according to the MAP score.

| Mean Average Precision (MAP) | | |
|---|---|---|
| | AIDS | Mutag |
| $T = 3, K = 5$ | 0.847 | 0.858 |
| $T = 3, K = 10$ | 0.865 | 0.871 |

| Inference time (in ms) | | |
|---|---|---|
| | AIDS | Mutag |
| $T = 3, K = 5$ | 64.39 | 66.03 |
| $T = 3, K = 10$ | 88.59 | 90.76 |

Table 22: MAP and inference time trade-off of variants of multi-round lazy IsoNet++ (Edge) with fixed $T$. Rows colored green indicate the best $K$ according to the MAP score.

| Mean Average Precision (MAP) | | |
|---|---|---|
| | AIDS | Mutag |
| $T = 3, K = 5$ | 0.847 | 0.858 |
| $T = 4, K = 5$ | 0.881 | 0.887 |
| $T = 5, K = 5$ | 0.886 | 0.909 |

| Inference time (in ms) | | |
|---|---|---|
| | AIDS | Mutag |
| $T = 3, K = 5$ | 64.39 | 66.03 |
| $T = 4, K = 5$ | 85.02 | 87.33 |
| $T = 5, K = 5$ | 106.24 | 109.1 |

Table 23: MAP and inference time trade-off of variants of multi-round lazy IsoNet++ (Edge) with fixed $K$. Rows colored green and yellow indicate the best and second best $T$ according to the MAP score.

| Mean Average Precision (MAP) | | |
| --- | --- | --- |
| | AIDS | Mutag |
| $K = 5$ | 0.795 | 0.805 |
| $K = 6$ | 0.828 | 0.837 |
| $K = 7$ | 0.852 | 0.848 |
| $K = 8$ | 0.862 | 0.851 |

| Inference time (in ms) | | |
| --- | --- | --- |
| | AIDS | Mutag |
| $K = 5$ | 72.63 | 73.46 |
| $K = 6$ | 86.03 | 87.77 |
| $K = 7$ | 100.26 | 102.6 |
| $K = 8$ | 114.33 | 115.01 |

Table 24: MAP and inference time trade-off of variants of multi-layer eager IsoNet++ (Edge) with increasing $K$. Rows colored green and yellow indicate the best and second best $K$ according to the MAP score.

| Mean Average Precision (MAP) | | | | | | |
| --- | --- | --- | --- | --- | --- | --- |
| | AIDS | Mutag | FM | FR | MM | MR |
| $K = 5$ | 0.622 | 0.710 | 0.730 | 0.662 | 0.655 | 0.708 |
| $K = 8$ | 0.641 | 0.731 | 0.745 | 0.701 | 0.658 | 0.711 |
| $K = 10$ | 0.679 | 0.736 | 0.741 | 0.712 | 0.691 | 0.74 |
| $K = 12$ | 0.651 | 0.728 | 0.743 | 0.697 | 0.687 | 0.699 |

| Inference time (in ms) | | | | | | |
| --- | --- | --- | --- | --- | --- | --- |
| | AIDS | Mutag | FM | FR | MM | MR |
| $K = 5$ | 52.94 | 53.16 | 53.23 | 53.12 | 53.32 | 53.34 |
| $K = 8$ | 83.97 | 84.47 | 84.64 | 84.38 | 85.41 | 84.51 |
| $K = 10$ | 104.87 | 105.21 | 105.72 | 105.33 | 105.66 | 105.73 |
| $K = 12$ | 125.99 | 126.33 | 126.53 | 126.39 | 126.79 | 126.59 |

Table 25: MAP and inference time trade-off of variants of GMN with increasing $K$. Rows colored green and yellow indicate the best and second best $K$ according to the MAP score.

| | AIDS | Inference time (in ms) |
| --- | --- | --- |
| $K = 5$ | 0.69 | 19.77 |
| $K = 6$ | 0.717 | 20.83 |
| $K = 7$ | 0.697 | 21.96 |
| $K = 8$ | 0.709 | 23.02 |

Table 26: MAP and inference time trade-off of variants of IsoNet (Edge) with increasing $K$. Rows colored green and yellow indicate the best and second best $T$ according to the MAP score.

## G.7 Contribution of refining alignment matrix in inference time

In GMN, computing the embeddings of nodes after the message passing step at each layer dominates the inference time. However, in the case of IsoNet++ models, we observe the refinement of the alignment matrix at each layer or round to also be time-intensive. In table 27, we show the contribution of embedding computation and matrix updates to the total inference time. The updates to $P$ constitute the largest share of inference time for multi-layer variants. This can be attributed to the refinement of $P$ after every message passing step, equaling the frequency of embedding computation. In the case of multi-round variants, both embedding computation and updates to $P$ contribute almost equally since $P$ is refined only at the end of each round, after several layers of message passing alongwith embedding computation.

| Models | Embedding Computation | Matrix Updates |
|---|---|---|
| multi-layer IsoNet++ (Node) | 13.7 | 68.3 |
| multi-layer IsoNet++ (Edge) | 19.7 | 76.3 |
| multi-round IsoNet++ (Node) | 34.1 | 47.8 |
| multi-round IsoNet++ (Edge) | 54.9 | 39.9 |

Table 27: Inference time contribution of embedding computation and matrix updates by multi-layer and multi-round IsoNet++ (Node) and IsoNet++ (Edge) models.

## G.8 Transfer ability of learned models

In this section, we evaluate the transfer ability of each trained model across datasets. In table 28, we report the Mean Average Precision (MAP) scores for models trained using the AIDS and Mutag datasets respectively evaluated on all six datasets. We observe that despite a zero-shot transfer from one of the datasets to all others, variants of IsoNet++ show the best accuracy.

| Test across other datasets when trained on **AIDS** | | | | | | |
|---|---|---|---|---|---|---|
| | AIDS | Mutag | FM | FR | MM | MR |
| GraphSim [2] | 0.356 | 0.225 | 0.192 | 0.198 | 0.210 | 0.215 |
| GOTSim [11] | 0.324 | 0.275 | 0.370 | 0.339 | 0.314 | 0.361 |
| SimGNN [1] | 0.341 | 0.264 | 0.374 | 0.344 | 0.331 | 0.383 |
| EGSC [31] | 505 | 0.255 | 0.473 | 0.451 | 0.447 | 0.499 |
| H2MN [45] | 0.267 | 0.272 | 0.319 | 0.281 | 0.262 | 0.297 |
| Neuromatch [23] | 0.489 | 0.287 | 0.442 | 0.403 | 0.386 | 0.431 |
| GREED [32] | 0.472 | 0.307 | 0.477 | 0.452 | 0.436 | 0.490 |
| GEN [22] | 0.557 | 0.291 | 0.445 | 0.427 | 0.437 | 0.496 |
| GMN [22] | 0.622 | 0.342 | 0.569 | 0.544 | 0.532 | 0.588 |
| IsoNet (Node) [35] | 0.659 | 0.459 | 0.612 | 0.562 | 0.588 | 0.640 |
| IsoNet (Edge) [35] | 0.690 | 0.468 | 0.620 | 0.568 | 0.624 | 0.627 |
| multi-layer IsoNet++ (Node) | 0.756 | 0.685 | 0.825 | 0.767 | 0.781 | 0.794 |
| multi-layer IsoNet++ (Edge) | 0.795 | 0.683 | 0.800 | 0.751 | 0.792 | 0.785 |
| multi-round IsoNet++ (Node) | 0.825 | 0.702 | 0.828 | 0.777 | 0.800 | 0.825 |
| multi-round IsoNet++ (Edge) | 0.847 | 0.741 | 0.846 | 0.799 | 0.833 | 0.836 |
| Test across other datasets when trained on **Mutag** | | | | | | |
| | AIDS | Mutag | FM | FR | MM | MR |
| GraphSim [2] | 0.188 | 0.472 | 0.190 | 0.193 | 0.205 | 0.198 |
| GOTSim [11] | 0.194 | 0.272 | 0.185 | 0.192 | 0.202 | 0.182 |
| SimGNN [1] | 0.206 | 0.283 | 0.203 | 0.209 | 0.220 | 0.195 |
| EGSC [31] | 0.296 | 0.476 | 0.391 | 0.333 | 0.309 | 0.355 |
| H2MN [45] | 0.209 | 0.276 | 0.204 | 0.207 | 0.223 | 0.197 |
| Neuromatch [23] | 0.275 | 0.576 | 0.368 | 0.304 | 0.304 | 0.325 |
| GREED [32] | 0.328 | 0.567 | 0.388 | 0.335 | 0.356 | 0.370 |
| GEN [22] | 0.278 | 0.605 | 0.359 | 0.308 | 0.312 | 0.330 |
| GMN [22] | 0.299 | 0.710 | 0.434 | 0.361 | 0.389 | 0.394 |
| IsoNet (Node) [35] | 0.458 | 0.697 | 0.503 | 0.456 | 0.446 | 0.486 |
| IsoNet (Edge) [35] | 0.472 | 0.706 | 0.499 | 0.438 | 0.467 | 0.489 |
| multi-layer IsoNet++ (Node) | 0.601 | 0.810 | 0.695 | 0.611 | 0.628 | 0.614 |
| multi-layer IsoNet++ (Edge) | 0.527 | 0.805 | 0.558 | 0.507 | 0.560 | 0.563 |
| multi-round IsoNet++ (Node) | 0.645 | 0.851 | 0.679 | 0.626 | 0.652 | 0.655 |
| multi-round IsoNet++ (Edge) | 0.625 | 0.858 | 0.639 | 0.598 | 0.634 | 0.650 |

Table 28: Test MAP of all graph retrieval methods on different datasets, when they were trained on **AIDS** (top half) and **Mutag** (bottom half) dataset. The numbers with green and yellow indicate the best, second best method respectively.

