# OpenReview forum: "Iteratively Refined Early Interaction Alignment for Subgraph Matching based Graph Retrieval"
_NeurIPS.cc/2024/Conference — NeurIPS 2024 poster_

### Official Review · Reviewer_WLV9 · 2024-07-03

**Soundness:** 3
**Presentation:** 3
**Contribution:** 3
**Rating:** 7
**Confidence:** 3

**Summary:**

The paper introduces EINSMATCH, an early interaction graph neural network for subgraph matching and retrieval that maintains and refines explicit alignments between query and corpus graphs. EINSMATCH has several novelties: computing embeddings guided by injective alignments between graphs, updating alignments lazily over multiple rounds, and using node-pair partner interactions. The model learns to identify alignments between graphs despite only having access to pairwise preferences during training, without explicit alignments. Experiments on several datasets show that EINSMATCH significantly outperforms existing methods.

**Strengths:**

1. The analysis of GMN’s worse performance compared to IsoNet (in Section 1) is intriguing.
2. Extensive experimental results are presented with ample baselines and std errors included as well for most numbers reported.
3. Source code with data splits is released for ease of reproducing the results.
4. A large amount of ablation and parameter sensitivity studies are performed.

**Weaknesses:**

1. The datasets used in the experiments are relatively small, with at most ~20 nodes.
2. It would be interesting to see the transfer ability of the learned model across datasets, e.g. training on PTC and testing on AIDS, to see how the proposed model would handle such challenging but realistic scenarios.

**Questions:**

N/A

---

> ### Author Rebuttal · Authors · 2024-08-07
>
> We thank the reviewer for their insightful review. The weaknesses raised in the review are addressed below.
>
> > The datasets used in the experiments are relatively small, with at most ~20 nodes.
>
> In early experiments, we found relatively small size of query graphs to actually present more challenge to the learner. Given our distant supervision, there are a larger number of promising alignments between query and corpus graphs that may lead to the relevant supervision.
>
> During the rebuttal, we performed experiments with graph pairs generated from the Cox2 dataset.  Here, we considered 100 query graphs with $|V|\in[25,35]$, and 500 corpus graphs with $|V|\in[40,50]$. The following table shows the results.
>
> | Models                   | Test mAP score on Cox2 |
> |--------------------------|-------|
> | NeuroMatch | 0.5036 |
> | EGSC | 0.4727 |
> | GREED | 0.1874 |
> | GEN | 0.6590 |
> | GMN | 0.6818 |
> | IsoNet (Node) | 0.7641 |
> | IsoNet (Edge) | 0.7356 |
> | EinsM. (Node) Lazy | 0.7291 |
> | EinsM. (Node) Eager | 0.7393 |
> | EinsM. (Edge) Lazy | **0.8302** |
> | EinsM. (Edge) Eager | 0.7949 |
>
> We observe that our method EinsMatch (Edge) Lazy outperforms the baselines significantly. The node variant of our model is marginally outperformed by IsoNet (Node).
>
> > It would be interesting to see the transfer ability of the learned model across datasets, e.g. training on PTC and testing on AIDS, to see how the proposed model would handle such challenging but realistic scenarios.
>
> During the rebuttal period, we performed experiments evaluating the transfer ability of the learned models across datasets. In particular, we chose the weights for each model trained using the AIDS and Mutag datasets respectively and evaluated them on all six datasets. The results are shown below. We observe that despite a zero-shot transfer from one of the datasets to all others, variants of EinsMatch show the best accuracy.
>
> **Models Trained on AIDS**
>
> | Test Datasets $\to$                   | AIDS  | Mutag |   FM   |   FR   |   MM   |   MR   |
> |--------------------------|-------|-------|--------|--------|--------|--------|
> | GraphSim                 | 0.356 | 0.225 | 0.192  | 0.198  | 0.210  | 0.215  |
> | GOTSim                   | 0.324 | 0.275 | 0.370  | 0.339  | 0.314  | 0.361  |
> | SimGNN                   | 0.341 | 0.264 | 0.374  | 0.344  | 0.331  | 0.383  |
> | EGSC                     | 0.505 | 0.255 | 0.473  | 0.451  | 0.447  | 0.499  |
> | H2MN                     | 0.267 | 0.272 | 0.319  | 0.281  | 0.262  | 0.297  |
> | NeuroMatch               | 0.489 | 0.287 | 0.442  | 0.403  | 0.386  | 0.431  |
> | GREED                    | 0.472 | 0.307 | 0.477  | 0.452  | 0.436  | 0.49   |
> | GEN                      | 0.557 | 0.291 | 0.445  | 0.427  | 0.437  | 0.496  |
> | GMN                      | 0.622 | 0.342 | 0.569  | 0.544  | 0.532  | 0.588  |
> | IsoNet (Node)            | 0.659 | 0.459 | 0.612  | 0.562  | 0.588  | 0.640  |
> | IsoNet (Edge)            | 0.690 | 0.468 | 0.620  | 0.568  | 0.624  | 0.627  |
> | EinsM. (Node) Multi Layer| 0.756 | 0.685 | 0.825  | 0.767  | 0.781  | 0.794  |
> | EinsM. (Edge) Multi Layer| 0.795 | 0.683 | 0.800  | 0.751  | 0.792  | 0.785  |
> | EinsM. (Node) Multi Round| 0.825 | 0.702 | 0.828  | 0.777  | 0.800  | 0.825  |
> | EinsM. (Edge) Multi Round| **0.847** | **0.741** | **0.846**  | **0.799**  | **0.833**  | **0.836** |
>
> **Models Trained on Mutag**
>
> |  Test Datasets $\to$                   | AIDS  | Mutag |   FM   |   FR   |   MM   |   MR   |
> |--------------------------|-------|-------|--------|--------|--------|--------|
> | GraphSim                 | 0.188 | 0.472 | 0.190  | 0.193  | 0.205  | 0.198  |
> | GOTSim                   | 0.194 | 0.272 | 0.185  | 0.192  | 0.202  | 0.182  |
> | SimGNN                   | 0.206 | 0.283 | 0.203  | 0.209  | 0.220  | 0.195  |
> | EGSC                     | 0.296 | 0.476 | 0.391  | 0.333  | 0.309  | 0.355  |
> | H2MN                     | 0.209 | 0.276 | 0.204  | 0.207  | 0.223  | 0.197  |
> | NeuroMatch               | 0.275 | 0.576 | 0.368  | 0.304  | 0.304  | 0.325  |
> | GREED                    | 0.328 | 0.567 | 0.388  | 0.335  | 0.356  | 0.37   |
> | GEN                      | 0.278 | 0.605 | 0.359  | 0.308  | 0.312  | 0.33   |
> | GMN                      | 0.299 | 0.71  | 0.434  | 0.361  | 0.389  | 0.394  |
> | IsoNet (Node)            | 0.458 | 0.697 | 0.503  | 0.456  | 0.446  | 0.486  |
> | IsoNet (Edge)            | 0.472 | 0.706 | 0.499  | 0.438  | 0.467  | 0.489  |
> | EinsM. (Node) Multi Layer| 0.601 | 0.810 | **0.695**  | 0.611  | 0.628  | 0.614  |
> | EinsM. (Edge) Multi Layer| 0.527 | 0.805 | 0.558  | 0.507  | 0.560  | 0.563  |
> | EinsM. (Node) Multi Round| **0.645** | 0.851 | 0.679  | **0.626**  | **0.652**  | **0.655**  |
> | EinsM. (Edge) Multi Round| 0.625 | **0.858** | 0.639  | 0.598  | 0.634  | 0.650  |

---

> > ### Comment · Reviewer_WLV9 · 2024-08-08
> >
> > Thank you for your rebuttal.

---

### Official Review · Reviewer_65Ze · 2024-07-14

**Soundness:** 3
**Presentation:** 3
**Contribution:** 3
**Rating:** 6
**Confidence:** 3

**Summary:**

This work proposes a GNN architecture for graph matching problem. The innovations in the work mainly lie in the early interaction between the query graph and the corpus graph, and the node-pair matching setting, as well as the lazy update technique in each round. Empirical results show that this work significantly outperforms the baselines on all the datasets.

**Strengths:**

- The writing of the paper is clear and good.
- The methodology is solid, and the illustration is informative.
- The results are very promising, both in predictive performance and runtime. And the ablation experiments are well-designed.

**Weaknesses:**

- Some background part is not well explained, for example the graph matching problem, and the optimal transport problem. Readers may need external resources to understand them. Of course it is hard to explain everything in detail in the main paper.

**Questions:**

- Any related work indicating early interaction is good? As mentioned in line 36.
- In line 124 the notation $A_q \leq PA_c P^T$ is a bit confusing.
- In equation 5, what is the sign of $\tau$? If it's positive then you're encouraging the entropy to be higher, which means the entries of P are more uniform? That is a bit counter-intuitive.
- What is the complexity of Sinkhorn Knopp algorithm and Gumbel Sinkhorn?
- Is the K-layer GNN in each round t shared?
- Any intuition why lazy updates works better? Is it easier to optimize or other aspects?
- I don't quite understand why the work is not differentiable on nodes with different classes, as mentioned in appendix A.
- Is there other measure between the $P$ and $P^*$? I feel $Tr(P^TP^*)$ does not capture all the information of the matrices.

**Limitations:**

As mentioned by the authors, the quality of the matching relies on the distance function, which is a general problem for graph matching works. Also I am a little dubious about the time complexity of the P matrix update, in the experiments it is more expensive than the baseline GMN.

---

> ### Author Rebuttal · Authors · 2024-08-07
>
> We thank the reviewer for their insightful feedback. The questions raised in the review are addressed below.
> > Any related work indicating early interaction is good?
>
> The GMN paper itself provides some evidence: GMN-match (early cross-graph interaction) is slower but convincingly superior in terms of accuracy to GMN-embed (late interaction between whole-graph representations). Other domains provide strong supporting evidence [a,b]. E.g., in textual entailment (NLI) tasks [b], where we need to classify if sentence s1 implies sentence s2, late comparison of (say) transformer-based embeddings of the two sentences, computed separately, is convincingly beaten by injecting s1 concat s2 into a comparable transformer and fine-tuning a classification head for the NLI task.
>
> [a] A. Lai and J. Hockenmaier. Learning to predict denotational probabilities for modeling entailment.
>
> [b] Entailment as Few-Shot Learner.  Sinong Wang, Han Fang, Madian Khabsa, Hanzi Mao, Hao Ma. arXiv:2104.14690v1
>
> >  In line 124  $A _q\le PA _c P^{\top}$ is a bit confusing.
>
>
> By $A _q\le PA _c P^{\top}$, we mean the constraint holds true for each entry, i.e.,
> $A _q[u,v]\le (PA _c P^{\top})[u,v]$ for all u,v.
>
> Suppose  $A  _q$ is the node-node adjacency matrix of the query graph and   $A  _c$ is the node-node adjacency matrix of the corpus graph. Suppose, we could find a permutation $P$ of the node numbering (i.e., rows and columns) of $A  _c$, which is easily verified as $P A  _c P^T$, then $G _q\subseteq G _c$ implies that every edge in $A  _q$ will also be present in $P A _c P^T$.  In other words, if $A _q[u,v]=1$ then  $(P A _c P^T)[u,v]$ must also be 1. This leads to the stated inequality.
>
>
> > Eq 5: the sign of $\tau$
>
>  Yes, we indeed want to make the entropy of $P$ a bit higher as this allows us to convert the hard permutation matrix into a smooth doubly stochastic matrix. To gain basic intuition into this, consider the argmax to softmax conversion. The selection $\max _i a _i$, given a vector of real numbers $\vec{a}$, is rewritten as $\max _{\vec{z}\in \Delta} \vec{z} \cdot \vec{a}$, where $\Delta$ is the unit simplex ($\sum _i z _i =1$). Adding a maximizer of the entropy of $z$ readily yields the softmax function. Just like adding an entropy maximizer turns a vector of logits into a softmax distribution, adding an entropy maximizer turns a matrix of logits into a doubly stochastic soft permutation.  For technical details, see Cuturi’s classic paper and related works [10, 26].
>
> [10] M. Cuturi. Sinkhorn distances: Lightspeed computation of optimal transport. Advances in neural information processing systems, 26:2292–2300, 2013.
>
> [26]  G. Mena, D. Belanger, S. Linderman, and J. Snoek. Learning latent permutations with gumbel-sinkhorn networks. arXiv preprint arXiv:1802.08665, 2018.
>
> > What is the complexity of Sinkhorn Knopp algorithm and Gumbel Sinkhorn?
>
> As mentioned in Appendix D.5, both these algorithms consist of some number of alternating row and column scalings. If the input matrix is $N\times  N$, then each scaling takes $O(N^2)$ time. The number of scaling iterations is usually a small fixed constant (20 in our experiments).
>
> > Is the K-layer GNN in each round t shared?
>
> Yes.  We regard the same GNN as being trained after each refinement applied to the proposed (soft) permutation.
>
> > Any intuition why lazy updates works better?
>
>  Please refer to the global response.
>
> >why the work is not differentiable on nodes with different classes.
>
> We meant that, at present, EinsMatch is not coded to account for node or edge labels. Our matching process is purely structural. If there are hard constraints that red nodes in the query graph can match only red nodes in the corpus graph, or there is an additional cost in the form of the difference in their degree of redness, we need to investigate if Sinkhorn-style updates will survive zeroing out parts of $P$. The problem seems complicated enough for a separate piece of work.
>
>
> > Is there other measure between the P and P*?  I feel $Tr(P^\top P^*)$ does not capture all the information of the matrices.
>
> Indeed, other measures like Frobenius distance $\Vert P-P^*\Vert _F$ may be used. In Figure B in global-rebuttal-pdf, we updated the histograms, which reveal similar observations as with the Trace metric. Note that $\Vert P-P^*\Vert _F ^2 = 2|V| - 2\, Tr(P^TP^*)$ if P and P* are both hard permutation matrices, but not necessarily the same if they are soft permutation matrices.
>
> > Also I am a little dubious about the time complexity of the P matrix update, in the experiments it is more expensive than the baseline GMN.
>
> L716–L720 give an idea of the clock time needed to train EinsMatch variants. Appendix G.6 gives an idea of the inference times involved, particularly, contrasted against IsoNet (edge) — we find these times within the same order of magnitude (usually within 10% and 40% of each other). Figures 6 and 17 show that, at any fixed level of inference time investment, EinsMatch (lazy) has the most favorable MAP envelope.
>
> Following table shows the time splits between GNN embedding computation and matrix P updates, in percentage of time spent performing that step.
>
> | Models  | % Embedding Computation  | % Matrix $\textbf{P}$ updates |
> |--|--|--|
> | EinsMatch (Node) Multi Round | 34.1 | 47.8 |
> | EinsMatch (Node) Multi Layer | 13.7 | 68.3 |
> | EinsMatch (Edge) Multi Round | 54.9 | 39.9 |
> | EinsMatch (Edge) Multi Layer | 19.7 | 76.3 |
>
> These percentages don't add up to 100% because of contributions from other steps like node encoding and score computation.  The summary is that yes, updating  $P$ has a non-negligible but non-overwhelming cost, with the benefit of better accuracy.
>
> > As mentioned by the authors, the quality of the matching relies on the distance function.
>
> We absolutely agree with you. The same is the case for matching passages, images, video and audio. Good distance function design, mirroring human judgment, is critical to applications.

---

> > ### Comment · Reviewer_65Ze · 2024-08-11
> >
> > Thank you for the detailed reply and your effort. I would like to raise my score to 6.

---

### Official Review · Reviewer_1Kfh · 2024-07-16

**Soundness:** 3
**Presentation:** 3
**Contribution:** 3
**Rating:** 6
**Confidence:** 3

**Summary:**

This paper proposes an early interaction network for subgraph matching. The proposed method enables (1) early interaction GNNs with alignment refinement at both node and edge levels, where the alignments are refined by GNNs; (2) eager or lazy alignment updates,  and (3) node-pair partner interaction, instead of node partner interaction, which improves the model efficiency and embedding quality. The paper includes extensive experiments on real-world datasets, demonstrating the superiority and effectiveness of the proposed approach.

**Strengths:**

1. The paper is well-written and easy to understand. The notations are clearly clarified and explained.
2. The idea of iteratively early interaction between graphs for more accurate subgraph retrieval is novel and reasonable to me, which is also verified by the experimental results.
3. The performance superiority of the proposed method against other baselines is significant and promising.

**Weaknesses:**

1. Considering the high complexity of the proposed method, an ablation study (or a hyperparameter sensitivity study) would be beneficial to provide a clearer understanding of the success of the proposed method. In addition, how did the authors tune hyperparameters (apart from T and K) for their approach and the competitors?
2. Experiments are only conducted on small graph datasets (with an average of 11 nodes). It would be beneficial to discuss or evaluate the applicability of the proposed approach to larger graph datasets.

**Questions:**

- What is the motivation and intuition behind lazy updates? In my understanding, "lazy" means updating the alignment map after every K GNN layers, and "eager" means updating the alignment map and letting the map affect the embedding at each layer. Is there any explanation for the consistently better performance of "lazy" updates?
- What do the multiple points with the same color in Figure 6 represent? Do they indicate variants with different hyperparameters?

**Limitations:**

Yes. This paper discussed the limitations in terms of efficiency and

---

> ### Author Rebuttal · Authors · 2024-08-07
>
> We thank the reviewer for their constructive feedback. The questions raised in the review are addressed below.
>
> >*ablation study (or a hyperparameter sensitivity study).. how did the authors tune hyperparameters?*
>
> We used hyperparameters from the code of IsoNet, which, thanks to the authors, also made the baseline implementations public. In IsoNet's code, the authors suggest that they already tuned the margin in the ranking loss and other hyperparameters for all the baselines. Hence, we used them as-is. We did tune the margin in the ranking loss of three baselines (EGSC, GREED, H2MN) which are not included in IsoNet codebase. More details are in line 703 onwards in Appendix F.4. We run all methods for an unlimited number of epochs, with the patience parameter 50, i.e., training is stopped only if the performance on the validation set does not improve for 50 epochs.
>
> We would like to highlight that, for EinsMatch, we did not tune any hyperparameters beyond IsoNet. We performed a hyperparameter sensitivity study (not tuning) for $T$ and $K$, which shows that for wide ranges of $T$ and $K$, EinsMatch performs better than baselines. We report results for $T=3, K=5$ in the main table, because its running time is comparable to most baselines. Increasing $T$ and $K$ improves accuracy at more computation cost.
>
> We did perform ablations on the node pair partner interaction component in Table 4 and the effect of lazy updates in Table 3 in the main paper. We write the results here.
>
> The following table compares the performance of node partner interaction with EinsMatch, which involves node pair partner interaction. Node pair partner interaction consistently outperforms node partner interaction for both eager and lazy variants of EinsMatch (Node).
>
> |Model|AIDS|Mutag|FM|FR|MM|MR|
> |-----|----|-----|--|--|--|--|
> |Node partner (Lazy)|0.776|0.829|0.851|0.819|**0.844**|0.840|
> |Node pair partner (Lazy)|**0.825**|**0.851**|**0.888**|**0.855**|0.838|**0.874**|
> |Node Partner (Eager)|0.668|0.783|0.821|0.752|0.753|0.794|
> |Node pair partner (Eager)|**0.756**|**0.810**|**0.859**|**0.802**|**0.827**|**0.841**|
>
> The following table compares the performance of the eager and lazy variants of EinsMatch (Node) and EinsMatch (Edge). Lazy multi-round updates consistently outperform eager multi-layer updates across all datasets, for both Node and Edge models.
>
> |Model|AIDS|Mutag|FM|FR|MM|MR|
> |-----|----|-----|--|--|--|--|
> |EinsM. (Node, Eager)|0.756|0.810|0.859|0.802|0.827|0.841|
> |EinsM. (Node, Lazy)|**0.825**|**0.851**|**0.888**|**0.855**|**0.838**|**0.874**|
> |EinsM. (Edge, Eager)|0.795|0.805|0.883|0.812|0.862|0.886|
> |EinsM. (Edge, Eager)|**0.847**|**0.858**|**0.902**|**0.875**|**0.902**|**0.902**|
>
> We observe that our method, in the presence of the node pair partner interaction component, performs better and lazy update generally outperforms eager update.
>
> > applicability of the proposed approach to larger graph datasets.
>
> In early experiments, we found relatively small size of query graphs to actually present more challenge to the learner. Given our distant supervision, there are a larger number of promising alignments between query and corpus graphs that may lead to the relevant supervision.
>
> During the rebuttal, we performed experiments with graph pairs generated from the Cox2 dataset.  Here, we considered 100 query graphs with $|V|\in[25,35]$, and 500 corpus graphs with $|V|\in[40,50]$. The following table shows the results.
>
> | Models                   | Test mAP score on Cox2 |
> |--------------------------|-------|
> | NeuroMatch | 0.5036 |
> | EGSC | 0.4727 |
> | GREED | 0.1874 |
> | GEN | 0.6590 |
> | GMN | 0.6818 |
> | IsoNet (Node) | 0.7641 |
> | IsoNet (Edge) | 0.7356 |
> | EinsM. (Node) Lazy | 0.7291 |
> | EinsM. (Node) Eager | 0.7393 |
> | EinsM. (Edge) Lazy | **0.8302** |
> | EinsM. (Edge) Eager | 0.7949 |
>
> We observe that our method EinsMatch (Edge) Lazy outperforms the baselines significantly. The node variant of our model is marginally outperformed by IsoNet (Node).
>
> >  motivation and intuition behind lazy updates?
>
> Please refer to the global response.
>
> > What do the multiple points with the same color in Figure 6 represent?
>
>  We vary rounds \(T\) and layers \(K\). Each point corresponds to a $(T,K)$ combination.  The color, as the legend shows, corresponds to an algorithm family. The goal is to study the tradeoff between inference speed and ranking accuracy (MAP).

---

> > ### Comment · Reviewer_1Kfh · 2024-08-13
> > **Thanks for your reply.**
> >
> > Thanks for your response, which addressed my questions. I will maintain current score.

---

### Official Review · Reviewer_sihU · 2024-07-21

**Soundness:** 3
**Presentation:** 2
**Contribution:** 3
**Rating:** 6
**Confidence:** 3

**Summary:**

The paper proposes EinsMatch, a neural approach for graph retrieval, i.e. the problem of finding  the best candidates of a corpus of graphs that contain a subgraph isomorphic to a given query graph. EinsMatch is based on a Graph Neural Network architecture and introduces technical improvements over existing methods: (i) message-passing iterations across query and candidate graphs which are guided by an injective alignment; (ii) the iterations are lazy in the sense that multiple of them are computed with a fixed alignment before refining it; (iii) node pairs are considered as potential partners instead of single nodes in the message-passing steps. In addition, the authors propose a variant which works at the level of edges and natively leverages edge matchings. This variant seems to be the most competitive one in the experiments the authors run. In general, EinsMatch significantly outperforms a variety of other methods. The authors also report insightful ablation studies on the number of alignment rounds and message passing layers.

**Strengths:**

- The proposed architecture nicely interpolates between late and early aggregation approaches, whilst ensuring injectiveness of the candidate mapping. In this sense, it appears as a sensible generalisation of other approaches.

- The proposed model is highly performant if compared with other methods, and, in particular offers a competitive accuracy-inference time tradeoff if compared with other approaches as GMNs.

- The reported ablation studies are insightful in that they shed light on the impact of important hyperparameters, namely the number of layers and rounds.

**Weaknesses:**

- I found the paper somewhat hard to follow. The contributions are rather technical and only one single figure in the manuscript illustrate the approach. I think it would be particularly helpful if the authors improved on this aspect.

- Still on the presentation side, I found it hard to clearly understand the specific contributions w.r.t. previous works. I believe the paper would increase in quality if the authors would state these more clearly in the main corpus.

- The authors only provide a high-level intuition behind some specific technical contributions, e.g. the choice of working with node-pair partners. Would it be possible to refer to more formal arguments around those?

**Questions:**

Please see Weaknesses. Also:

- The readers refer oversmoothing problems in some of the previous methods. Can they expand on how their approach side-steps this?

- Can the authors better explain lines 333 — 336? I found them hard to understand.

- How does the computational complexity compare with that of IsoNet?

**Limitations:**

The authors do not seem to explicitly discuss them in the main corpus, but mention two aspects in Appendix A.

---

> ### Author Rebuttal · Authors · 2024-08-07
>
> We thank the reviewer for their detailed and insightful review.
>
> > *paper hard to follow...only one single figure*
>
> In the global rebuttal PDF, we include another diagram (Figure A) to distinguish between node-pair interaction (earlier work) vs node-pair partner interaction, and also contrast them with IsoNet. If our paper gets accepted, we will include this diagram in the paper.
>
> > *contributions w.r.t. previous works.*
>
> Early interaction models are conventionally more powerful in other domains [a,b], yet in the context of subgraph retrieval, GMN (early interaction) is outperformed by IsoNet (late interaction).
>
> [a] A. Lai and J. Hockenmaier. Learning to predict denotational probabilities for modeling entailment.
>
> [b] Entailment as Few-Shot Learner.  S Wang, H Fang, M Khabsa, H Mao, Hao Ma. arXiv:2104.14690v1
>
> In this work, we build a new early interaction model, which addresses the limitations of prior work, also using lessons from IsoNet and GMN. Here, we further elaborate on our contributions.
>
> 1. GMN does not use any explicit alignment structure. They use a non-injective attention mechanism, which changes from layer to layer. In our work, we model inter-graph alignments as relaxed bijective permutation maps that are *proposed*, then $K$ layers of GNN *score* the proposal, leading to a *refined* proposal. Thus, the node embeddings of one graph become dependent on the paired graph, implementing early interaction. To the best of our knowledge, this is the first early interaction model for subgraph matching where proposed alignments and GNN layers help refine each other.
> 1. When we compute the message of  node-pair $(u,v)$ in $G _q$, we also use $(u',v')$ in $G _c$ which match with $(u,v)$. Therefore, for $u\in G  _q$, the node embedding $\mathbf{h} ^{(q)}  _u$ not only depends on nodes in $G _c$ that align with $u$, but also on those nodes in $G _c$, which are aligned with the $\text{Nbr}(u)$ neighbors of $u$. In terms of a formal argument, we have:
> $$\mathbf{h} ^{(q)}  _u = f(\mathbf{h} ^{(q)}  _u, \\{ \mathbf{h} ^{(q)}  _v: v\in  \text{Nbr}(u) \\}, \\{\mathbf{h} ^{(c)}  _ {v'} : v' \in G _c \text{  aligns with some  } v \in \text{Nbr}(u)\\})\ \ \ (1)$$
> Existing early interaction models like GMN  do not capture signals from $\\{v' \in G _c \text{  aligns with some  } v \in \text{Nbr}(u)\\}$, they only capture signal from $u' \in G _c$ that matches with $u$.  Hence, in GMN,  we have:
> $$\mathbf{h} ^{(q)}  _u = f(\mathbf{h} ^{(q)}  _u, \\{\mathbf{h} ^{(q)}  _v: v\in  \text{Nbr}(u)\\}, \\{\mathbf{h} ^{(c)}  _ {u'} : u' \in G _c \text{  aligns with } u\\})\ \ \ (2)$$
> As our experiments suggest, approach (1) gives better performance than (2).  We provide a more detailed description of this in lines 146-161 in the main paper.
> 1. We propose two protocols for updating node embeddings and alignments (Lazy and Eager). In Lazy update, we first perform $K$ GNN runs followed by then one single alignment update and in Eager, we update the alignment in each of the $K$ message passing steps.
>
> We will include these elaborations in the additional page, if our paper gets accepted.
>
>
> >  how their approach side-steps oversmoothing? explain lines 333 — 336.
>
> The usually symmetric and parameter-free aggregation of messages from a node’s neighbors is known to weaken graph representations computed by GNNs [c]. L334 points out that scaling $h(u’)$ with a relaxed (evolving) permutation $P[u,u’]$ breaks the symmetry and conjectures that this reduces the oversmoothing effect, through task-driven cross-attention between two graphs (as against self-attention in GAT [d]).
>
> [c] A Survey on Oversmoothing in Graph Neural Networks; T. K. Rusch, M. M. Bronstein, S Mishra.
>
> [d] Graph attention networks, P Veličković, G Cucurull, A Casanova, A Romero, P Liò, Y Bengio.
>
>
> > How does the computational complexity compare with that of IsoNet?
>
> Let's consider three node models — IsoNet (Node), Eager EinsMatch (Node) and Lazy EinsMatch (Node) with $T$ rounds, each with $K$ propagation steps.
>
> **IsoNet (Node)**
>
> Initial node featurization takes $O(N)$ time. Node representation computation aggregates node embeddings over all neighbors, leading to complexity of $O(E)$ for each message passing step and computation of $P$ takes $O(N^2)$ time. The overall complexity for all the steps becomes $O(N^2+KE)$ where $K$ is the number of message passing steps.
>
> **Eager Multi-layer EinsMatch (Node)**
>
> 1. Initial Node featurization takes $O(N)$ time
> 1. EinsMatch first computes intermediate embeddings $z$ (Eq. 33, Page 16), which admits complexity of $O(N)$ for each layer per node, since it computes $\sum _{u'\in V _c} h^{(c)} _{k}(u') \textbf{P} _{k}[u,u']$ for each node in each layer. The total complexity for $K$ steps is $O(KN^2)$.
> 1. Next we  compute (Eq. 34) $h _{k+1}$ which gathers messages $z$ from all neighbors per node per layer. The total complexity contributed by this equation is $O(KE)$.
> 1. Next we update $P _k$  for each layer which has complexity of $O(KN^2)$.
>
> Hence, the total complexity is $O(KN^2+KE+KN^2)=O(KN^2)$.
>
> **Lazy Multi-round EinsMatch (Node)**
>
> Here, the key difference from Eager version is that the doubly stochastic matrix $P _t$ from round $t$ is used to compute $z$ and the $K$-step-GNN runs in $T$ rounds. This increases the complexity of step 2 and 3 to $O(KTN^2+KTE)$. Matrix $P _t$ is updated a total of $T$ times, which changes the complexity of step 4 to $O(TN^2)$. Hence, the total complexity is $O(KTN^2+TN^2+KTE) = O(KTN^2)$.
>
> Hence, complexity of IsoNet is $O(N^2+KE)$. EinsMatch-Eager has complexity $O(KN^2)$ and EinsMatch-Lazy has complexity $O(KTN^2)$. However, this increased complexity comes with the benefit of significant acuracy boost, as our experiments suggest.

---

> > ### Comment · Reviewer_sihU · 2024-08-11
> >
> > I acknowledge reading the authors' rebuttal, which I indeed found very helpful in better appreciating their contributions.
> >
> > I strongly believe that the additional figures they proposed, the point-by-point discussion of contributions wrt previous works, as well as the complexity analyses should be included in the next revision of the manuscript.
> >
> > I also appreciate the clarification on oversmoothing. I believe that, should the authors have space, they could also make these passages less cryptic in the paper. After all, they are mostly (interesting) conjectures.

---

### Author Rebuttal · Authors · 2024-08-07

We are thankful to all the reviewers for taking out time to provide us with detailed reviews. Through this rebuttal, we aim to address questions that came up across multiple reviews. Figures have also been shared through the rebuttal PDF attached herewith.

> Why is Lazy update better? (Reviewers 1Kfh, 65Ze)

In L125-142 and L164-L167, we provide justification for lazy updates.
The underlying optimal alignment between the graphs remains the same for each of $K$ steps (layers) of message passing of GNN. Hence, when GNN performs its operation, the alignment proposal (matrix $P$) should be fixed. In eager update, this matrix keeps changing across the message passing steps, whereas in lazy update, $P$ is updated (and improved) only after $K$ layers of GNN, which enhances the inductive bias.

Elaborating, as we discussed in L125-142, we start with a combinatorial formulation and its solution. Define,
$$\text{cost}(P; A _q, A _c)   =
\sum _{u\in [n],v\in[n]}  [\big(A _q-P A _c P^{\top}\big) _+] [u,v]$$

This cost is minimized through Gromov Wasserstein (GW) approach, where P is updated as
$$    P _{t} \leftarrow  \text{argmin} _{P} \textrm{Trace}\left(P^T\nabla _{P}\ \text{cost}(P;A _q,A _c) \big| _{P = P _{t-1}}\right) + \tau \sum _{u,v}P[u,v] \cdot \log P[u,v] .
$$

One can view each GW update step as a lazy update step. In the above, $P _{t-1}$ is used to compute the cost and in our method, we use it to compute the GNN embeddings and effectively those embeddings can approximate the $P _{t-1}$  dependent cost. Then, $P _t$ is updated by solving an entropy-regularized OT problem and we update $P _{t}$ using Sinkhorn iterations, which effectively optimizes entropy-regularized GNN guided cost.

---

### Decision · Program_Chairs · 2024-09-25

**Decision:**

Accept (poster)

**Comment:**

The paper proposes a GNN-based neural architecture for graph retrieval, i.e., finding the best candidates for a set of graphs containing a subgraph isomorphic to a given query graph. The architecture proposes several innovations, i.e., message-passing iterations are guided by an injective alignment; a lazy update technique is used in each round by computing with a fixed alignment before refining it; and pairs of vertices are considered instead of single vertices in the message-passing steps. Moreover, the authors propose an extension to edge-level matching. The paper is complemented with a rather extensive empirical study, suggesting that the proposed architecture outperforms competing approaches.

After an extensive rebuttal by the authors, all reviewers vote to accept the present paper. I follow this assessment.